# Engineering ZrO$_2$–Ru interface to boost Fischer-Tropsch synthesis to olefins

Hailing Yu [1,2,6], Caiqi Wang [1,6], Xin Xin[1,2,6], Yao Wei[2,3], Shenggang Li [1,4] ✉, Yunlei An[1], Fanfei Sun[5], Tiejun Lin [1] ✉ & Liangshu Zhong [1,4] ✉

Understanding the structures and reaction mechanisms of interfacial active sites in the Fisher-Tropsch synthesis reaction is highly desirable but challenging. Herein, we show that the ZrO$_2$-Ru interface could be engineered by loading the ZrO$_2$ promoter onto silica-supported Ru nanoparticles (ZrRu/SiO$_2$), achieving 7.6 times higher intrinsic activity and ~45% reduction in the apparent activation energy compared with the unpromoted Ru/SiO$_2$ catalyst. Various characterizations and theoretical calculations reveal that the highly dispersed ZrO$_2$ promoter strongly binds the Ru nanoparticles to form the Zr-O-Ru interfacial structure, which strengthens the hydrogen spillover effect and serves as a reservoir for active H species by forming Zr-OH* species. In particular, the formation of the Zr-O-Ru interface and presence of the hydroxyl species alter the H-assisted CO dissociation route from the formyl (HCO*) pathway to the hydroxy-methylidyne (COH*) pathway, significantly lowering the energy barrier of rate-limiting CO dissociation step and greatly increasing the reactivity. This investigation deepens our understanding of the metal-promoter interaction, and provides an effective strategy to design efficient industrial Fisher-Tropsch synthesis catalysts.

Fischer-Tropsch synthesis (FTS) is a versatile platform technology to produce various fuels and value-added chemicals from oil-alternative carbon resources (e.g., coal, natural gas, biomass, solid waste, and CO$_2$) via a syngas (CO + H$_2$)-mediated chemical conversion process[1,2]. Compared with the traditional commercial FTS technology with liquid fuels as the target products, the production of olefins, especially long-chain olefins (C$_{5+}^=$), has received great attention due to the wide application of olefins for synthesizing surfactants, polymers, high-quality lubricants, and biodegradable detergents[3–6]. In the past decade, milestone achievements have been made in direct syngas conversion to olefins, which fall into two categories: the bifunctional catalysis route using oxide-zeolite catalysts and the Fischer-Tropsch to olefins (FTO) route using metal carbide catalysts[7–14]. Nonetheless, both routes still suffer from high water

gas shift (WGS) activity, leading to low carbon utilization efficiency and low olefin yield.

Recently, much attention has been paid on modified metal-based catalysts featuring low intrinsic WGS reactivity for the FTO reaction[15–18]. For example, Xie et al. reported that Na/S/Mn-modified hexagonal close-packed (hcp) Co could achieve 54% selectivity to lower olefins with selectivities to CH$_4$ and CO$_2$ below 17% and 3%, respectively, at 1% CO conversion[16]. In addition, Yu et al. revealed that Na-promoted metallic Ru nanoparticles (NPs) exhibited 80.2% olefins selectivity with an ultralow combined selectivity for CH$_4$ and CO$_2$ (<5%) at 45.8% CO conversion[17]. However, compared with non-noble metal catalysts such as Fe- or Co-based catalysts, the cost of the noble metal Ru greatly limits its application despite its excellent performance for syngas conversion. Therefore, it is vital to greatly reduce the Ru loading and

$^1$CAS Key Laboratory of Low-Carbon Conversion Science and Engineering, Shanghai Advanced Research Institute, Chinese Academy of Sciences, Shanghai 201210, PR China. $^2$University of Chinese Academy of Sciences, Beijing 100049, PR China. $^3$Shanghai Institute of Applied Physics, Chinese Academy of Sciences, Shanghai 201800, PR China. $^4$School of Physical Science and Technology, ShanghaiTech University, Shanghai 201210, PR China. $^5$Shanghai Synchrotron Radiation Facility, Shanghai Advanced Research Institute, Chinese Academy of Sciences, Shanghai 201210, PR China. $^6$These authors contributed equally: Hailing Yu, Caiqi Wang, Xin Xin. ✉e-mail: lisg@sari.ac.cn; lintj@sari.ac.cn; zhongls@sari.ac.cn

develop more efficient and cost-effective catalysts to achieve higher Ru utilization efficiency and higher intrinsic activity for the FTO reaction.

Interface engineering has emerged as an effective strategy to tune the catalytic behavior of FTS. By constructing abundant interfacial sites on metal nanoparticles, e.g., metal-oxide sites[19–22], single atom-decorated surface sites[23–25], the geometric and electronic properties as well as the coordination environment of metal NPs could be substantially changed, leading to much-improved intrinsic activity and product selectivity. These newly formed interfacial active sites can facilitate the activation of the reactants and promote the formation and transformation of the intermediates. For instance, the Co–Zr interface has been successfully engineered to promote CO adsorption and dissociation, thereby enhancing the reactivity of Co-based FTS catalysts. Liu et al. suggested that the Co–ZrO$_2$ interface, featuring single-site dispersion of ZrO$_2$ on surfaces of Co nanoparticles, enhanced both H$_2$ adsorption and CO dissociation[24]. Li et al. reported that the Ru$_1$Zr$_1$/Co catalyst with dual atomic sites of Ru and Zr on surfaces of Co nanoparticles effectively weakened the C–O bond and promoted C–C coupling[25]. However, the role of interfacial active sites in alternating the reaction pathway and the evolution of activated CO species has been rarely reported.

In this work, the ZrO$_2$-doping strategy is used to successfully engineer silica-supported Ru nanoparticles to form the Zr–O–Ru interfacial structure (ZrRu/SiO$_2$), which shows 7.6 times increase in the Ru-normalized intrinsic activity and a 4.1-fold increase in the space-time-yield of olefin products. The experimental apparent activation energy also decreases by approximately 45% compared to the unpromoted Ru/SiO$_2$ catalyst. Combining performance evaluation experiments, catalyst characterizations, and density functional theory (DFT) calculations, the role of the unique Zr–O–Ru interfacial structure is revealed. It is found that introducing the ZrO$_2$ promoter greatly improves hydrogen spillover and migration, and changes the H-assisted CO dissociation pathway, leading to much enhanced catalytic activity and Ru utilization efficiency.

## Results

### CO hydrogenation performance

We previously demonstrated that Ru/SiO$_2$ catalysts with 5 wt.% Ru loading showed excellent catalytic activity for the FTO reaction[17]. Here, the Ru loading was further decreased to 2 wt.%, and the ZrO$_2$ promoter was introduced onto Ru/SiO$_2$ (abbreviated as xZr/Ru, where x denotes the Zr/Ru molar ratio) to form ZrO$_2$–Ru interfacial sites. The xZr/Ru catalysts were evaluated under the same reaction conditions of 260 °C, H$_2$/CO = 2, 5 bar, and 3000 mL·g$^{-1}$·h$^{-1}$, as shown in Supplementary Fig. 1. Additionally, the intrinsic activity was assessed under the similar CO conversion level (7.2–10.7%) by changing the weight hourly space velocity (Fig. 1). The CO conversion and ruthenium-weight-based activity (ruthenium time yield, RuTY) strongly depended on the Zr/Ru molar ratio, and a volcano-like trend was observed when increasing the ZrO$_2$ content. As the Zr/Ru molar ratio increased from 0 to 0.5, CO conversion dramatically increased from 7.2% to 40.2% (Fig. 1a). The 0.5Zr/Ru sample exhibited the highest RuTY value of

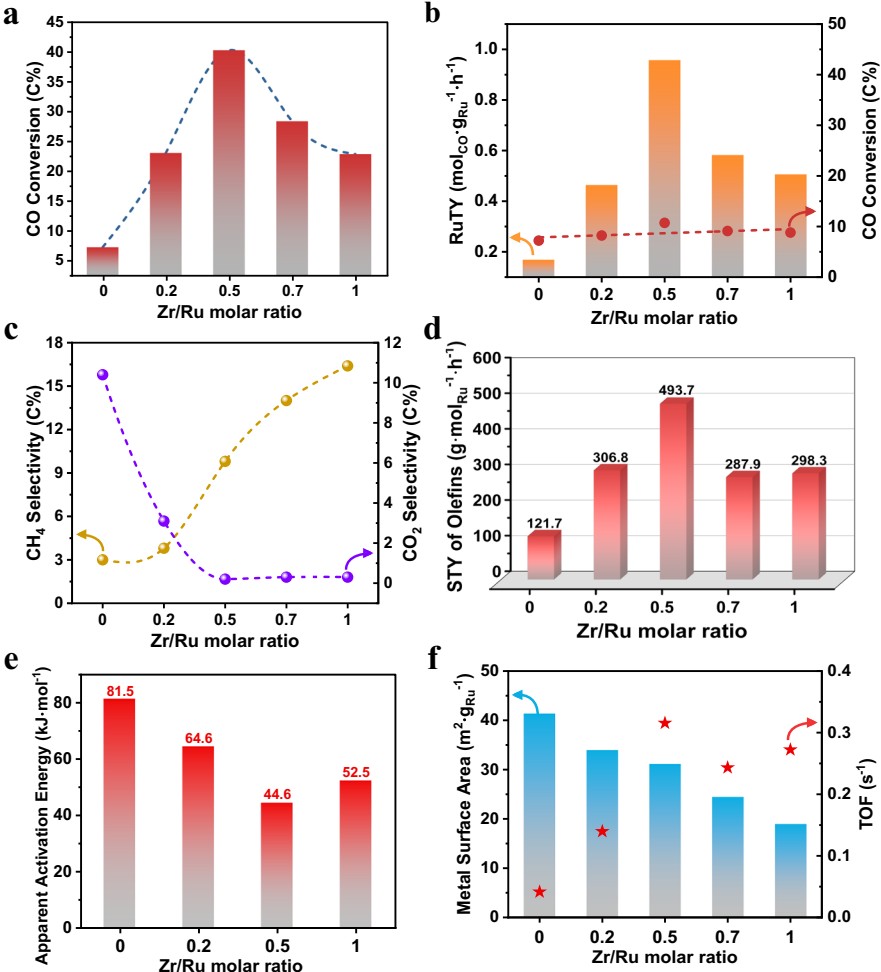

**Fig. 1 | CO hydrogenation performance. a, b** CO conversion and RuTY, **c** product selectivity, and **d** space-time yield of olefins for ZrO$_2$-promoted SiO$_2$-supported ruthenium catalysts. **e** Apparent activation energies for CO conversion determined by the Arrhenius plots shown in Supplementary Fig. 5. **f** Metallic Ru surface areas (columns) determined by CO chemisorption and TOF (pentagram) plotted against the Zr/Ru molar ratios.

0.955 $mol_{CO} \cdot g_{Ru}^{-1} \cdot h^{-1}$ and the TOF value of 0.316 s⁻¹, which were 5.7 and 7.6 times higher than that of the unpromoted sample (0.167 $mol_{CO} \cdot g_{Ru}^{-1} \cdot h^{-1}$ and 0.041 s⁻¹), respectively (Fig. 1b, f). However, further increase in the $ZrO_2$ content caused the catalytic activity to decrease, resulting in a RuTY value of 0.504 $mol_{CO} \cdot g_{Ru}^{-1} \cdot h^{-1}$ for 1Zr/Ru, which was still 3.0-fold higher than that of the 0Zr/Ru catalyst (Fig. 1b). In comparison, the as-obtained 0.5Zr/Ru presents superior FTS intrinsic activity to most previously reported Ru-based catalysts (Supplementary Table 1). Moreover, $ZrO_2$ doping significantly suppressed $CO_2$ production while increasing $CH_4$ selectivity (Fig. 1c). Although a slight decrease in olefins selectivity was observed as increasing Zr/Ru molar ratio, the space-time yield (STY) of olefins gradually increased and reached a maximum value of 493.7 $g \cdot mol_{Ru}^{-1} \cdot h^{-1}$ for 0.5Zr/Ru, which was 4.1-fold higher than that of 0Zr/Ru (Fig. 1d). As the Zr/Ru molar ratio further increased, the STY of olefins decreased to 287.9 $g \cdot mol_{Ru}^{-1} \cdot h^{-1}$ for 0.7Zr/Ru and 298.3 $g \cdot mol_{Ru}^{-1} \cdot h^{-1}$ for 1Zr/Ru, which remained far higher than that of 0Zr/Ru. For comparison, the STY of olefins for 0.5Zr/Ru catalyst was even higher than that of 5%Ru/SiO₂ with a 5 wt.% Ru loading under similar CO conversion level (Supplementary Fig. 2). Performance of 0.5Zr/Ru catalyst remained stable within 110 h of the test (Supplementary Fig. 3). CO conversion and RuTY remained at the average values of 38.2% and 0.914 $mol_{CO} \cdot g_{Ru}^{-1} \cdot h^{-1}$, respectively. In addition, $CO_2$ selectivity remained consistently below 0.5%, while $CH_4$ selectivity was kept between 5.8% and 7.0%, resulting in $C_{5+}$ product selectivity of approximately 80%.

$ZrO_2$ introduction also influenced the product distribution. A volcanic trend was observed for the olefins distribution and the chain-growth probability (α) of hydrocarbons (Supplementary Fig. 4) when increasing the Zr/Ru molar ratio. The largest fraction of long-chain olefins ($C_{5+}^=$) reached 76.7%, which was much higher than that of 0Zr/Ru (57.2%). The α value for 0.5Zr/Ru showed a 1.5-fold increase compared to that of 0Zr/Ru. These results suggest that adding an appropriate amount of $ZrO_2$ could greatly promote the carbon chain growth.

Given the substantial increase in the TOF for the $ZrO_2$-promoted Ru-based catalysts, the apparent activation energy ($E_a$) was further measured via kinetics experiments (Fig. 1e, Supplementary Fig. 5). As expected, $E_a$ was greatly reduced from 81.5 kJ/mol for 0Zr/Ru to 44.6 kJ/mol for 0.5Zr/Ru, showing a 45% drop. Although a slight increase in $E_a$ was observed when further increasing the Zr/Ru ratio, the $E_a$ value was still lower than that of 0Zr/Ru. These results suggest the possible formation of new and more efficient active centers for the $ZrO_2$-promoted Ru-based catalysts, which significantly lower the energy barrier for the rate-limiting CO dissociation step.

To exclude the effect of sodium and demonstrate the promoting effect of $ZrO_2$ promoter on Ru-based catalysts in the traditional Fischer-Tropsch synthesis process, we prepared and evaluated the catalytic performance of Na-free Ru/SiO₂−0Na and Zr−Ru/SiO₂−0Na. As shown in Supplementary Fig. 6, it was found that the addition of Zr promoter notably enhanced the catalytic activity, as evidenced by the increase of CO conversion from 24.5% for Ru/SiO₂−0Na to 45.4% for Zr−Ru/SiO₂−0Na. The above experimental results suggest that the promotional effect of Zr on supported Ru-based catalysts with or without the Na promoter are similar. Moreover, powder X-ray diffraction (XRD) patterns and transmission electron microscopy (TEM) images present similar metallic Ru phases without the $ZrO_2$ phase and similar particle sizes of metallic Ru nanoparticles (Supplementary Figs. 7, 8). In this work, we focus on the promotional effect of Zr promoter on Fischer-Tropsch synthesis to olefins, therefore the promotional effect of $ZrO_2$ will be studied in detail over xZr/Ru catalysts.

## Identification of the Zr−O−Ru interfacial structure

Several advanced characterization methods were employed to reveal the detailed catalyst structure and disclose the nature of the active sites responsible for the much-improved FTO intrinsic activity. A series of xZr/Ru samples with similar Ru contents close to the nominal value of 2 wt.% were synthesized and studied (Supplementary Table 2). Adding the $ZrO_2$ promoter had no significant influence on the textural properties of the catalysts (Supplementary Table 3 and Supplementary Fig. 9a). According to the powder XRD patterns (Supplementary Fig. 9b, c, Fig. 2a), $RuO_2$ (JCPDS, 43-1027) was reduced by $H_2$ to metallic Ru (JCPDS, 06-0663), and the metallic state of Ru was retained after the FTO reaction. No diffraction peaks for the crystalline phases of the $ZrO_2$ promoter were observed, indicating that the Zr species was well

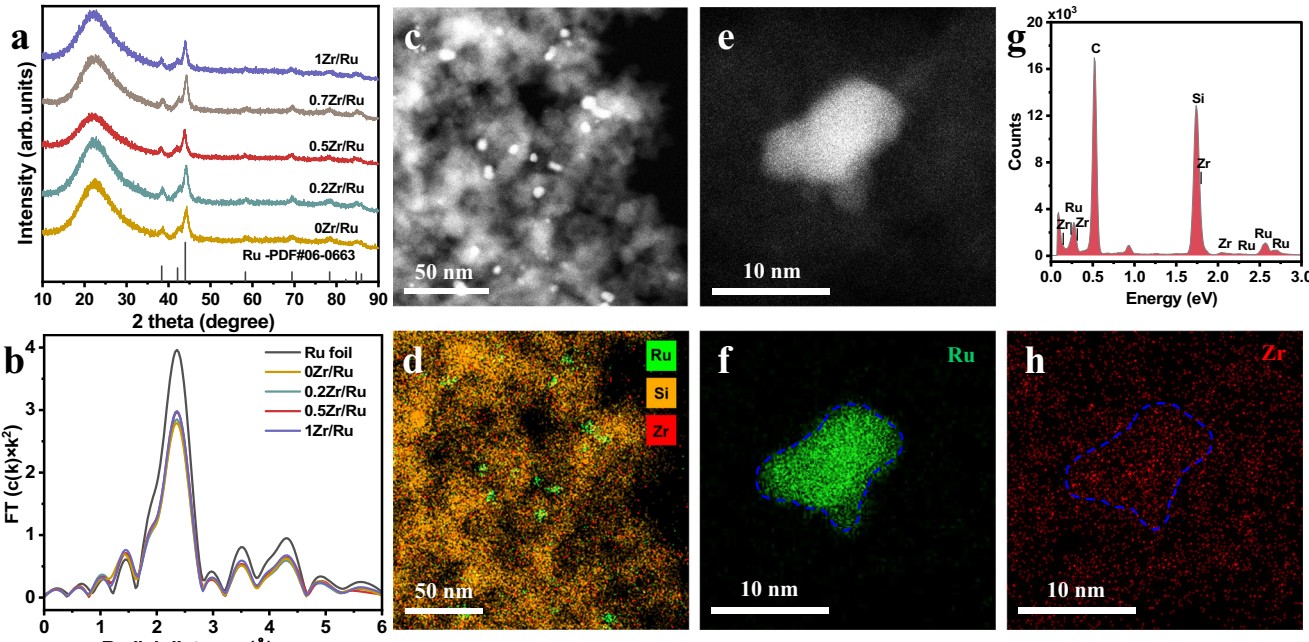

**Fig. 2 | Structural characterization of various xZr/Ru catalysts. a** XRD patterns of the reduced catalysts; **b** Ru K-edge EXAFS spectra of the reduced xZr/Ru samples; **c, d** HAADF-STEM images and STEM-EDS elemental mappings of the reduced 0.5Zr/Ru; **e–h** AC-HAADF-STEM images and elemental mappings of the reduced 0.5Zr/Ru sample. (Ru, Si, and Zr atoms are depicted in green, yellow, and red, respectively).

dispersed or in the amorphous phase. High-resolution TEM images of the reduced and spent xZr/Ru catalysts presented lattice fringes with a spacing of 2.03 Å assigned to the (101) plane of the hcp metallic Ru phase (Supplementary Figs. 10 and 11). Comparison of the average particle sizes of metallic Ru nanoparticles measured by TEM for all reduced and spent samples indicates that the $ZrO_2$ loading had little influence on the particle size of metallic Ru NPs (Supplementary Table 4).

Nevertheless, introducing the $ZrO_2$ promoter substantially affects the exposed Ru metallic surface area (MSA), which is considered to significantly influence CO dissociation and intrinsic reactivity[26]. The relationship between the Ru MSA determined by CO chemisorption and the intrinsic reactivity for all xZr/Ru catalysts was investigated (Fig. 1f). We found that the Ru MSA gradually decreased as the $ZrO_2$ content increased. A higher MSA usually means more Ru active sites for FTO, thus possibly leading to superior catalytic activity. However, the highest RuTY and TOF values were obtained over the 0.5Zr/Ru catalyst, indicating that the MSA is not the only key factor influencing the catalytic performance.

High-angle annular dark-field scanning transmission electron microscopy (HAADF-STEM) and energy dispersive X-ray (EDX) mapping characterizations (Fig. 2c, d and Supplementary Figs. 12 and 13) were employed to investigate the Ru and Zr dispersions on $SiO_2$. As shown in the HAADF-STEM images, Ru nanoparticles were clearly observed on the xZr/Ru catalysts. In addition, Ru nanoparticles, represented by green dots, were well-dispersed over the $SiO_2$ support as shown in the STEM-EDS elemental mapping. The Zr species, represented by red dots, were also highly dispersed over the entire sample. The AC-HAADF-STEM images and elemental mappings (Fig. 2e-h) further confirmed high dispersion of the Zr species on both the Ru NPs and supports without aggregation. As presented in Supplementary Fig. 14, after the reaction the Zr species also displayed no evident aggregation and remained highly dispersed on both the Ru NPs and supports. Besides, the elemental profiles of the STEM-EDS line-scanning further confirmed the presence of the Zr species over the Ru NPs.

We further investigated the local environment of the Ru sites in the Zr-promoted catalysts by X-ray absorption spectroscopy (XAS). According to the XANES spectra at the Ru K-edge, the Ru near-edge of the reduced samples almost coincides with that of Ru foil, suggesting the existence of metallic Ru sites (Supplementary Fig. 15). The coordination environment of the Ru sites was revealed by the R-space data of the $k^2$-weighted Ru K-edge FT-EXAFS results (Fig. 2b). For all reduced samples, only a major peak corresponding to the Ru–Ru coordination at ~2.4 Å related to the Ru foil was observed, consistent with the presence of the metallic Ru phase. Thus, no changes occurred to the metallic Ru phase after introducing the $ZrO_2$ promoter. The Ru K-edge FT-EXAFS spectra of the spent 0.5Zr/Ru also exhibited the metallic Ru phase (Supplementary Fig. 16a).

Elucidating the structure and chemical state of the Zr promoter is essential to understand its role in the xZr/Ru catalysts. The electronic and local structural information of the $ZrO_2$ promoter was obtained from X-ray photoelectron spectroscopy (XPS) and XAS measurements. From the XPS binding energy of the Zr $3d_{5/2}$ peak for the reduced xZr/Ru catalysts, the oxidation state of the Zr species was found to be +4 (Fig. 3a)[27]. Furthermore, the peak intensity was greatly enhanced as the Zr/Ru molar ratio increased. The XANES spectra at the Zr K-edge for the $ZrO_2$-promoted catalysts measured after the reduction are depicted in Fig. 3b, together with those of Zr foil and monoclinic $ZrO_2$ as references. The absorption edge energy corresponds to the typical value of $Zr^{4+}$ for all samples, and the white-line peak was split into two features: A and A' (Supplementary Fig. 17a). Moreover, as the Zr/Ru molar ratio increased, the intensity of peak A gradually increased, while the intensity of peak A' gradually decreased. The XANES spectra of the catalyst with a higher Zr/Ru molar ratio (1Zr/Ru) tended to match that of the monoclinic $ZrO_2$ standard, while those of the sample

with a lower Zr/Ru molar ratio (0.2Zr/Ru) exhibited a more prominent A' feature, corresponding the tetragonal $ZrO_2$[28]. This implies that the Zr species exist in the tetragonal crystal structure at the low loading, possibly due to Zr being atomically (or monolayer) dispersed over the $SiO_2$ and metallic Ru surfaces, while the monoclinic $ZrO_2$ structure emerges at the higher Zr loading[29]. Chemical states of the Ru and Zr species were found to remain unchanged after the reaction (Supplementary Fig. 18). Additionally, the Zr/Ru molar ratio of the spent 0.5Zr/Ru was calculated to be 1.28, close to that of the reduced 0.5Zr/Ru before the reaction. This further confirms that no significant changes occurred to the dispersion of $ZrO_2$ and metallic Ru during the reaction.

The experimental and best-fit FT-EXAFS spectra of the reduced xZr/Ru samples at the Zr K-edge are displayed in Fig. 3c. The shaded region in both spectra displays two prominent peaks, representing the scattering paths of the first and second shells. The peaks observed in the reference material were identified as the Zr–O ($R = 1.58$ Å) and Zr–Zr ($R = 3.10$ Å) scattering paths of $ZrO_2$. According to the Zr K-edge EXAFS fitting results, the Zr–O and Zr–Ru coordination numbers (CNs) decreased gradually as the $ZrO_2$ loading increased (Supplementary Table 5). Such changes in the coordination numbers evidence the transformation of the tetragonal $ZrO_2$ to the monoclinic $ZrO_2$, which is induced by the increasing $ZrO_2$ loading. In the case of the Zr-promoted catalyst, the first peak in the spectra corresponds to the Zr–O scattering path, but the second shell peak is shifted to ~2.7 Å, which differs from that of the Zr–Zr scattering path (~2.8 Å) of Zr foil (Supplementary Fig. 17b). By substituting the Ru atom with the Zr atom in calculating the scattering paths, a satisfactory fit was achieved. This suggests the possible intermixing between the Zr promoter and the Ru phase resulting in Zr–Ru scattering contributions, implying an intimate interaction between the metal-oxide-promoter ($ZrO_2$) and the active metal site (metallic Ru) in forming the Zr–O–Ru interfacial structures. Furthermore, the appearance of the Zr–Ru peak implies a reduced number of neighboring Zr atoms in this shell compared to bulk $ZrO_2$, indicating that the Zr atoms in the promoted samples are likely to be highly dispersed. The Zr–Ru scattering contributions were also observed in the Zr K-edge EXAFS results of the spent 0.5Zr/Ru catalyst, further demonstrating the strong interaction between the $ZrO_2$ promoter and metallic Ru resulting in the stable Zr–O–Ru interfacial structures during the reaction (Supplementary Fig. 16b).

From the XPS spectra for Ru 3d and Ru 3p shown in Fig. 3d and Supplementary Fig. 19, the binding energies of the Ru $3d_{5/2}$ peak and the Ru $3d_{3/2}$ peak were measured to be 279.9–280.2 eV and 461.0–461.5 eV, respectively, suggesting that Ru is in the metallic state. The relative Zr/Ru ratio was calculated from the ratio of the Zr $3d_{5/2}$ and Ru $3d_{5/2}$ XPS peak areas, which shows the relative concentrations of Zr and Ru on the catalyst surface (Fig. 3e). The Zr/Ru ratio measured by XPS increases considerably compared to the bulk value determined by X-ray fluorescence (XRF) analysis, suggesting that the Zr atoms are concentrated on the surface of the Ru NPs. This agrees well with the AC-HAADF-STEM observations shown in Fig. 2e-h and the decrease in the metallic Ru surface area (Fig. 1f). Additionally, Ar-ion sputtering was conducted to strip different depths of the 0.5Zr/Ru catalyst in the XPS measurements (Supplementary Fig. 20). The relative Zr/Ru molar ratio decreases from 1.57 to 0.67 as the etching time increases from 0 s to 30 s, which is closer to the nominal value (0.5) of the sample. These observations align with the results from the (AC)HAADF-STEM-EDS and XAS characterizations, further indicating that the $ZrO_2$ layer may be homogenously dispersed on the surface of the 0.5Zr/Ru catalyst. Apparently, a moderate loading of the $ZrO_2$ promoter can lead to the formation of a $ZrO_2$ monolayer over the catalyst surface, which benefits the formation of substantial Zr–O–Ru interface structures. However, based on the CO chemisorption results (Supplementary Table 4 and Fig. 3f), an excessive amount of $ZrO_2$ would cover the metallic Ru surface, thus hindering the access of the reactant molecules to the active sites and decreasing the catalytic activity.

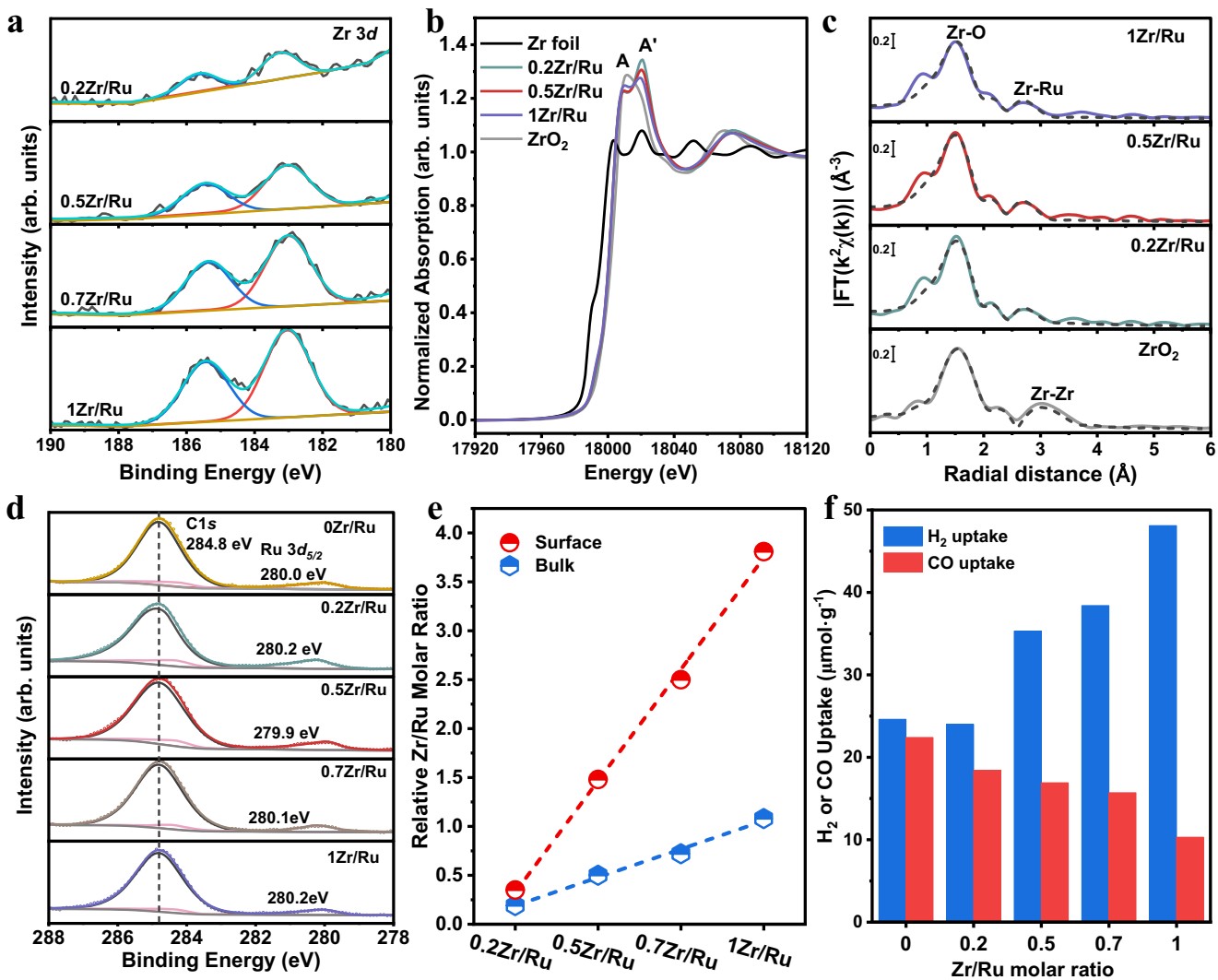

**Fig. 3 | Determination of the ZrO₂ promoter phase and Zr−O−Ru interfacial structure. a** XPS spectra of the Zr 3*d* line. **b** Stacked representation of Zr K-edge XANES. **c** Experimental (colored lines) and best fit (dashed black lines) FT-EXAFS spectra for samples with different Zr/Ru molar ratios measured at the Zr K-edge. **d** XPS spectra of various reduced catalysts at the Ru 3*d* level with Ru 3*d* and C1*s* contributions. **e** The molar Zr/Ru ratios in the bulk determined by XRF and at the surface calculated from XPS for various xZr/Ru catalysts. **f** Chemisorption uptakes of H₂ and CO over various xZr/Ru catalysts.

## Understanding the enhanced activity of Zr−O−Ru interfacial sites

To explore the influence of ZrO₂ addition on CO dissociation on the Zr−O−Ru interfacial sites, the temperature-dependent evolution of the CH₄ signal (m/z = 16) was monitored by carbon monoxide temperature programmed surface reaction (CO-TPSR) experiments (Supplementary Fig. 21). The peak temperatures of CH₄ for the ZrO₂-promoted catalysts are significantly lower than those for the unpromoted catalyst (0Zr/Ru), indicating that the enhanced CO dissociation activity is probably due to formation of the Zr−O−Ru interfaces[30].

In situ, time-resolved diffuse reflectance infrared Fourier transform spectroscopy (DRIFTs) was also performed to monitor the dynamic evolution of CO hydrogenation (Fig. 4). In the spectra of the 0Zr/Ru and 0.5Zr/Ru catalysts, IR bands of gaseous CO were observed in the region of 2200–2050 cm⁻¹. The primary IR bands at 2028 cm⁻¹ and 2012 cm⁻¹ were assigned to linearly adsorbed CO species on metallic Ru sites. Twin-type and bridge-type adsorbed CO species were observed at 1916 cm⁻¹ and 1763 cm⁻¹, respectively[31]. Catalysts containing the Zr−O−Ru interfacial structures exhibit weaker twin-type and bridge-type CO adsorptions but stronger linear CO adsorption. The bands attributed to the C−H stretching vibrations of the CHₓ* (x = 1−3) species appeared in the range of 2962−2857 cm⁻¹, whose intensities were enhanced when

increasing the exposure time. By comparison, the peak intensities of linear CO adsorption and C−H bands increased noticeably for Zr-promoted Ru-based catalysts, suggesting the presence of Zr−O−Ru interfacial sites would accelerate CO dissociation[22]. In addition, the shift in product distribution towards higher-molecular-weight hydrocarbons may be attributed to the enhanced CO activation at Zr−O−Ru interfacial sites, resulting in higher coverages of CHₓ species for C−C coupling reactions. A promotional effect toward hydrogenation over Zr−O−Ru interfacial sites also caused a slight increase in CH₄ selectivity[20,32].

Considering the widely recognized H-assisted CO dissociation mechanism for metallic Co- or Ru-based FTS catalysts[33,34], the kinetic isotope effects (KIE) resulting from H₂/D₂ switching were measured during CO hydrogenation (Fig. 5a). An inverse KIE with the ratio of the reaction rates using H₂ and D₂ ($k_H/k_D$) of 0.45 was observed resulting from the compensating thermodynamic (H₂ dissociation to H*; H* addition to CO* species to form HCO* or COH*) and kinetic (H* reaction with HCO* or COH*) isotope effects[35,36]. The observed isotopic effect confirms the preference of H-assisted CO dissociation routes over the 0.5Zr/Ru catalyst. We thus hypothesize that the promoted CO dissociation may be linked to the reactivity of H species around the ZrO₂−Ru interfaces.

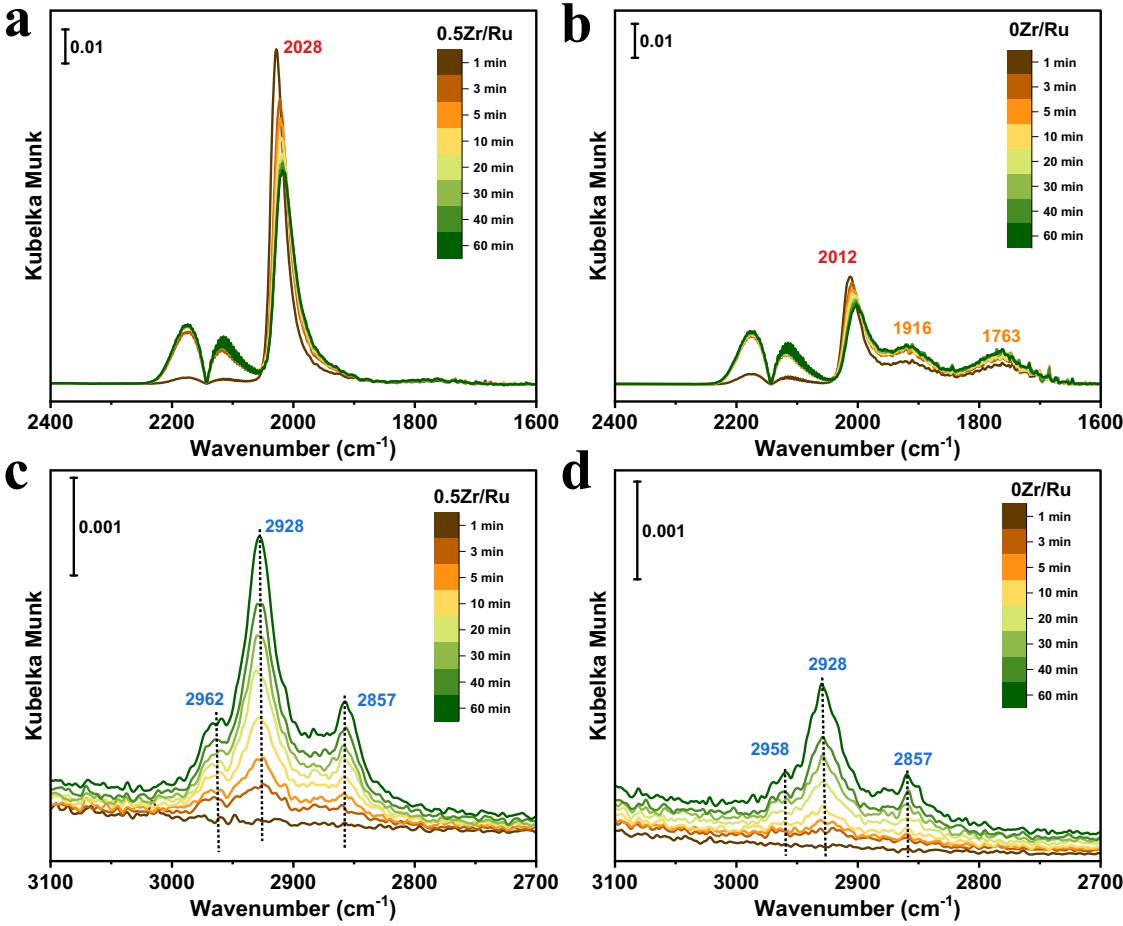

**Fig. 4 | CO hydrogenation mechanism revealed by in situ IR spectroscopy.** In situ time-resolved DRIFTs were obtained over the 0.5Zr/Ru (**a**, **c**) and 0Zr/Ru (**b**, **d**) catalysts at 260 °C upon exposure to syngas (CO: $H_2$: Ar = 1:2:7; 50 mL•min$^{-1}$).

Although the exposed metallic Ru surface areas decreased with the increasing Zr/Ru molar ratio, the increased $H_2$ uptake indicates that $ZrO_2$ promotion could greatly enhance $H_2$ chemisorption (Fig. 3f). This is further confirmed by the hydrogen temperature programmed desorption ($H_2$-TPD) results (Supplementary Fig. 22), and the increasing hydrogen coverage might be contributed to enhanced reactivity of the H species[37]. We also performed hydrogen temperature programmed reduction ($H_2$-TPR) experiments (Supplementary Fig. 23) and found that the reduction of $RuO_2$ to metallic Ru was enhanced for $ZrO_2$-doped Ru/SiO$_2$ catalysts, whose reduction temperatures were lower than that of the 0Zr/Ru sample, possibly due to the reduced metal-support interaction or the enhanced reactivity of the H* species upon $ZrO_2$ doping. To help understand the increasing hydrogen coverage for the $ZrO_2$-doped Ru catalysts, the hydrogen spillover effect was investigated by the hydrogen treatment of a mixture of the yellow $WO_3$ powder with the 0.5Zr/Ru and 0Zr/Ru catalysts. Blue $H_xWO_3$ usually forms during the hydrogen reduction of the yellow $WO_3$. As expected, darker blue was found for the mixture of the 0.5Zr/Ru and $WO_3$ after $H_2$ treatment, while light blue was observed for the mixture of the 0Zr/Ru and $WO_3$, implying that the incorporation of the $ZrO_2$ promoter enhanced the mobility of the hydrogen species (Supplementary Fig. 24)[38].

Faster hydrogen migration should lead to more rapid exchange of hydrogen species on the catalyst surface and $H_2$–$D_2$ exchange was further studied by in situ DRIFTs. As shown in Fig. 5b, after exposure of the 0.5Zr/Ru and 0Zr/Ru samples to $H_2$ at 50 °C for 30 min, an intense broad band centered at approximately 3400 cm$^{-1}$ appeared, which was assigned to the OH stretching vibration. When $D_2$ was introduced to the in situ chamber, two new vibrational bands at 2755 cm$^{-1}$ and 2633 cm$^{-1}$

were observed for 0Zr/Ru, which were correlated with new isolated and perturbed surface OD groups. Negative bands centered at approximately 3700 cm$^{-1}$ were observed, indicating the exchange of H with D in OH groups initially present on the silica surface to form Si-OD species[39]. Notably, in addition to the two peaks discussed above, a vibrational band was also observed at 2675 cm$^{-1}$ after switching to $D_2$ for the 0.5Zr/Ru sample, which was ascribed to multi-coordinated Zr–OD species[40–42]. The in situ DRIFTs for the $H_2$–$D_2$ exchange experiments confirmed that new Zr–OH groups were formed under the $H_2$ atmosphere on the $ZrO_2$-promoted catalyst surface. Furthermore, $H_2$–$D_2$ exchange experiments were also conducted to compare the $H_2$ migration rate and the $H_2$ storage capacity between the 0Zr/Ru and 0.5Zr/Ru catalysts (Fig. 5c). A sequence of gas switching, $H_2$→$D_2$→$H_2$, was performed, and HD molecules (m/z = 3) were detected in the mass spectra. The HD peak area of the 0.5Zr/Ru was 1.2 times that of the 0Zr/Ru, indicating that more H species were exchanged after $ZrO_2$ addition[43]. The number of surface hydroxyls was expected to increase with the addition of the $ZrO_2$ promoter due to the formation of Zr–OH groups, in line with the in situ DRIFTs for the $H_2$–$D_2$ exchange experiments. More importantly, these surface hydroxyls bound to the $ZrO_2$ promoter could easily be exchanged under hydrogen atmosphere, and the H* species would actively participate in the CO dissociation process (Fig. 5d)[44,45].

## Computational studies on reaction pathways over Zr–O–Ru interfacial sites

Density functional theory (DFT) calculations were further performed to reveal the possible pathway of hydrogen spillover over the $ZrO_2$-promoted Ru catalyst, which was modeled by placing a modest-sized

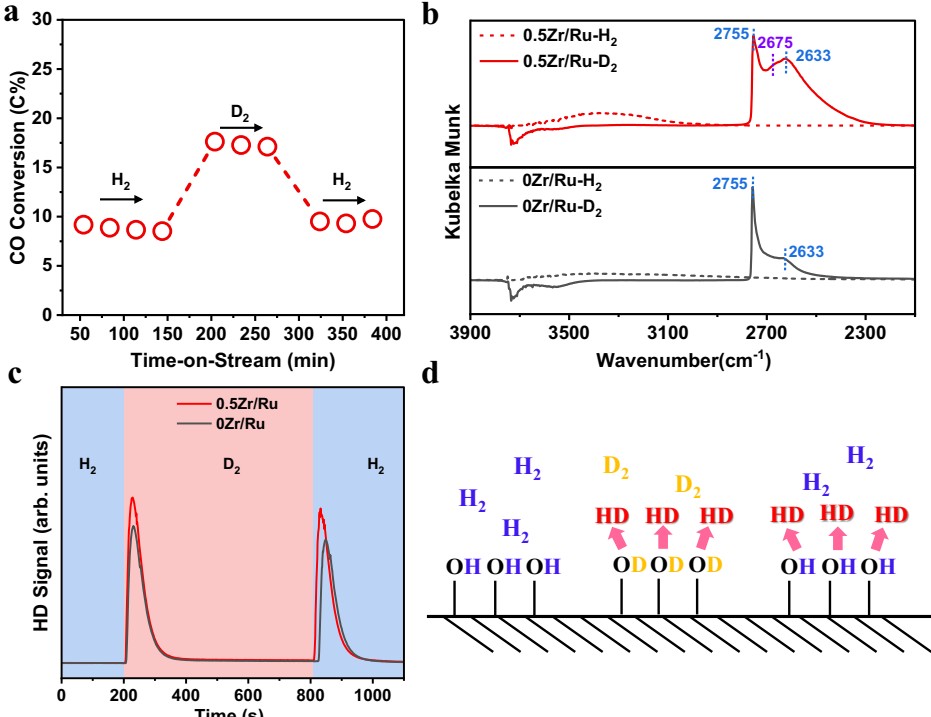

**Fig. 5 | Understanding the promotion effect of ZrO₂. a** The $H_2/D_2$ isotopic study of CO hydrogenation catalyzed by the 0.5Zr/Ru catalyst. **b** In situ DRIFTs corresponding to the $H_2$–$D_2$ exchange. The as-obtained 0.5Zr/Ru and 0Zr/Ru were introduced into an in situ chamber and treated with $H_2$ and $D_2$ under ambient pressure at 50 °C for 30 min before recording. **c** HD signal during a $H_2 \rightarrow D_2 \rightarrow H_2$ switch at room temperature over the 0.5Zr/Ru and 0Zr/Ru samples. **d** Schematic depiction of $H_2$–$D_2$ exchange experiments.

$(ZrO_2)_3$ cluster on the Ru (001) flat surface (Supplementary Fig. 25)[46,47]. As shown in the potential energy surface (Supplementary Fig. 26), the dissociation of molecular $H_2$ is an essentially barrierless step ($E_a = 0.02$ eV) with an exothermic dissociation energy of $E_{dis} = -0.77$ eV. Further migration of the H adsorbate (H*) from the Ru site to the O site on the $(ZrO_2)_3$ cluster is slightly endothermic by 0.44 eV with a modest energy barrier ($E_a = 0.97$ eV), so the reverse process is exothermic with a low energy barrier of 0.53 eV, and the Zr–O–Ru interfacial structure can promote the dissociation of molecular $H_2$ and the formation of surface hydroxyls from the diffusion of reactive H adsorbates. Our computational results are in line with our experimental observations, as our experiments indicate the strong interaction between the $ZrO_2$ promoter and Ru NPs and the formation of a unique $ZrO_2$–Ru interfacial structure, which not only facilitates the migration and exchange of the H species but also serves as an H species reservoir and increases the H species coverage near the active center by forming the Zr–OH species.

DFT calculations were further carried out to investigate the effect of the increased reactivity of the H species on CO dissociation[22]. Although CO adsorption on the Ru (001) and $(ZrO_2)_3$/Ru (001) surfaces are strong ($E_{ads} = -1.95$ eV and $-1.90$ eV, respectively), direct dissociation of the CO* adsorbate incurs energy barriers as high as 2.48 eV and 2.55 eV (Supplementary Fig. 27), so CO direct dissociation is kinetically prohibitively slow, consistent with the above KIE results. Under typical reaction conditions, $H_2$ activation is enhanced, and facile migration of the H* adsorbate after introducing the $ZrO_2$ promoter further results in hydroxylation of the $(ZrO_2)_3$ island and provides abundant Zr–OH species. Thus, a $Zr_3O_6H_5$ cluster was used to investigate the role of the Zr–OH species on CO dissociation under actual reaction conditions (Supplementary Fig. 28a). The potential energy surface for H-assisted CO dissociation over the Ru (001) and $Zr_3O_6H_5$/Ru (001) surfaces are further shown in Fig. 6 and Supplementary Table 6. Our calculations show that the Ru (001) and $Zr_3O_6H_5$/Ru (001) surfaces exhibit distinct

H-assisted CO activation mechanisms involving different intermediate species[48]. Over the Ru (001) surface, H-assisted CO dissociation via the formyl (HCO*) route ($E_a = 1.23$ eV) is much faster than that via the hydroxy-methylidyne (COH*) intermediates ($E_a = 1.74$ eV). Two additional steps are involved for C–O bond dissociation in the HCO* species, hydrogenation of the O atom in HCO* to form the hydroxymethylene (HCOH*) species with a relatively high energy barrier of $E_a = 1.22$ eV and cleavage of the C–O bond in the CHOH* species with a fairly low energy barrier of $E_a = 0.38$ eV to obtain CH* and OH*. In contrast, over the $Zr_3O_6H_5$/Ru (001) surface, CO dissociation is assisted by the hydroxy species formed at the Zr–O–Ru interfacial structure (i.e., Zr–OH*), which encounters a much lower energy barrier of 0.39 eV and occurs via the hydroxy-methylidyne (COH*) intermediate formed by hydrogen transfer to the O atom in CO*. Although the COH* species adsorbed at the Zr–O–Ru interfacial structure of the $ZrO_2$-promoted Ru surface can be further hydrogenated to the HCOH* species with a relatively high energy barrier of $E_a = 1.15$ eV, which is comparable to the highest energy barrier for the most favorable pathway of H-assisted CO dissociation on the Ru (001) surface of 1.23 eV, the COH* species can also dissociate with a much lower energy barrier of $E_a = 0.25$ eV (Supplementary Fig. 29). Thus, the $ZrO_2$-promoted Ru catalyst surface considerably accelerates CO dissociation via the COH* pathway assisted by the active Zr–OH* species. Our DFT calculations suggest that the OH* species are strongly adsorbed at the Zr sites with an adsorption energy of $E_{ads} = -1.33$ eV with gas phase $H_2O$ and $H_2$ as the reference states, much stronger than those adsorbed at the Ru sites with $E_{ads} = -0.26$ eV (Supplementary Table 7), thus the Zr atoms can stabilize the OH* species better than Ru atoms. The strong binding of OH* species at the Zr site significantly lowers the overall potential energy surface and makes the C–O bond dissociation kinetically and thermodynamically more favorable. On the Ru (001) surface, hydrogenation of OH* species is slightly endothermic by 0.39 eV with a high energy barrier of 1.22 eV, and further desorption of the

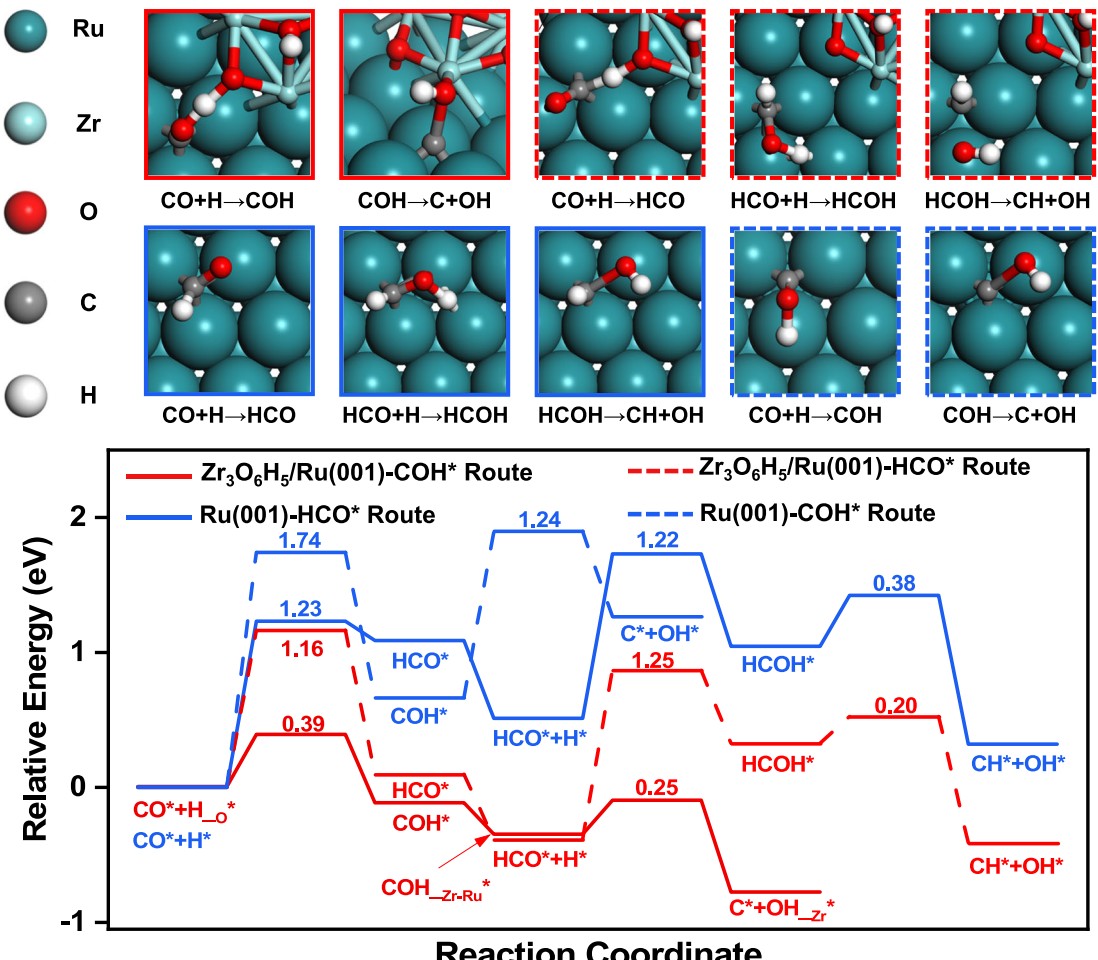

**Fig. 6 | DFT predictions of surface structures and reaction pathways.** Energy profiles and corresponding transition state structures of H-assisted CO dissociation via different intermediates (HCO* and COH*) over metallic Ru catalyst modeled by Ru (001) and the ZrO₂-promoted sample modeled by Zr₃O₆H₅/Ru (001). The adsorption sites other than the Ru site are indicated and further denoted by subscripts. Ru (dark green spheres), Zr (light green spheres), O (red spheres), H (white spheres), and C (gray spheres) atoms are shown.

adsorbed $H_2O$ molecules is modestly endothermic by 0.44 eV (Supplementary Fig. 30). In comparison, hydrogenation of OH* species adsorbed at the Zr sites encounters a slightly lower energy barrier of 1.13 eV, with an endothermicity of 0.81 eV, and further desorption of the adsorbed $H_2O$ molecules to recover the Zr sites are more endothermic on the $Zr_3O_6H_5$/Ru (001) surface ($E_{des} = 1.08$ eV), consistent with the high stability of the Zr–OH species on the ZrO₂-promoted Ru catalysts, although $H_2O$ desorption from the Zr site should readily occur considering the typical reaction temperatures.

Although the ZrO₂ promoter on the metallic Ru surface could better stabilize the H* adsorbates, the Ru surface may also have some adsorbed H* species, especially under high H coverage. The potential energy surface for H-assisted CO dissociation over the H-covered Ru catalyst model shows that increasing the H coverage could also promote the H-assisted CO dissociation for the preferred HCO* route (Supplementary Figs. 28b and 31), as the energy barriers for the three elementary steps, namely CO hydrogenation to HCO*, HCO* hydrogenation to HCOH*, and HCOH* dissociation, are mostly reduced from 1.23 eV, 1.22 eV, and 0.38 eV to 1.17 eV, 1.00 eV and 0.44 eV, respectively. DFT calculations were also performed using the $(ZrO_2)_3$/Ru (001) model, which can be considered to model the ZrO₂-promoted Ru catalyst under much lower $H_2$ pressure (Supplementary Figs. 25 and 32). The energy barrier for COH* formation assisted by the Zr–OH* species increases from 0.39 eV for the $Zr_3O_6H_5$/Ru (001) model to 0.54 eV for the $(ZrO_2)_3$/Ru (001) model, whereas that for COH*

dissociation to C* and OH* only slightly decreases from 0.25 eV to 0.23 eV, indicating the hydroxylated ZrO₂ promoter considerably enhances CO dissociation via the COH* pathway assisted by the Zr–OH* species under high H coverage.

These computational results thus show that the Zr–O–Ru interface facilitates a change in the H-assisted CO dissociation pathway from the HCO* pathway to the COH* pathway due to the presence of surface hydroxyl species. As the rate-limiting step of FTS is highly correlated with the energy barrier of C–O bond breaking, the formation of the Zr–O–Ru interfacial sites greatly promotes C–O bond splitting and accelerates the FTS reaction due to the change into the H-assisted CO dissociation route with a much lower energy barrier. This rationalizes the much higher FTS activity and space-time-yield of olefins products for our ZrRu/SiO₂ catalysts.

## Discussion

In summary, this work demonstrates that engineering ZrO₂–Ru interface sites effectively directs active hydrogen species to participate in CO activation and regulates the CO dissociation mechanism, achieving a higher intrinsic activity for the Fisher-Tropsch to olefins reaction. Compared with the unpromoted 0Zr/Ru sample, a 7.6-fold higher TOF for CO conversion (0.316 s⁻¹) and a 4.1-fold higher space-time-yield of olefins (493.7 g·mol$_{Ru}$⁻¹·h⁻¹) were obtained for 0.5Zr/Ru. The measured apparent activation energy also decreased by ~45% compared with the pure Ru/SiO₂. Various microscopy and spectral characterizations

suggest that the highly dispersed $ZrO_2$ species could strongly bind metallic Ru NPs to form Zr–O–Ru interface sites. Introducing the $ZrO_2$ promoter shows no obvious influence on the metallic Ru particle sizes but decreases the exposed Ru metallic surface area. However, the $ZrO_2$–Ru interface facilitates the mobility and exchange of surface H species due to the strengthened hydrogen spillover effect, which also serves as a reservoir for active H species by forming Zr–OH* species. Our computational studies further show that the presence of active hydroxyl species greatly increased the reactivity of H species around the Zr–O–Ru interfacial sites, thus altering the H-assisted CO dissociation route from the formyl (HCO*) to hydroxy-carbonyl (COH*) pathway. The changed reaction intermediates and CO dissociation route significantly reduce the rate-limiting barriers of C–O dissociation, leading to superior catalytic activity and increased Ru utilization efficiency. These results not only shed light on the understanding of the mechanism in the Fischer-Tropsch synthesis reaction but also open new avenues for designing novel more efficient catalysts for hydrogenation reactions.

## Methods

### Catalyst preparation

Ruthenium-based catalysts promoted by $ZrO_2$ promoters were prepared by incipient co-impregnation methods. The concentrations of impregnating solutions were calculated to obtain approximately 2.0 wt.% ruthenium in the final samples, the atomic ratio of Na to Ru was 0.5 and the atomic ratio of Zr promoter to Ru varied from 0 to 1. The samples were denoted as xZr/Ru (x = 0, 0.2, 0.5, 0.7, and 1.0), where x represents the Zr/Ru molar ratio. After impregnation, the as-prepared samples were dried at 60 °C for 4 h and calcinated in air at 400 °C for 3 h with a ramping rate of 2 °C/min. $Ru/SiO_2$–0Na and $Zr$–$Ru/SiO_2$–0Na catalysts without the Na promoter were also prepared by the impregnation method. The Ru loading was maintained at 2 wt.% with the same Zr/Ru molar ratio of 0.5.

### Catalyst evaluation

CO hydrogenation was performed using a fixed bed reactor to assess the effects of the Zr promotor on the catalytic activity and selectivity over Ru-based FTO catalysts. Each experiment started with loading 1 g of sample (40–60 mesh) diluted with 2–3 g $SiO_2$ into the reactor. The sample was reduced in pure $H_2$ flowing at 450 °C for 4 h with a heating rate of 2 °C/min. Prior to the CO hydrogenation reaction, the reactor temperature was cooled down to 180 °C, and then the reaction pressure was pressurized with the feed gas ($H_2$/CO = 2, including 3% $N_2$ as the inner standard) to reaction pressure, while the temperature was stepwise (2 °C/min) increased to reaction temperature. The reaction space velocity (3000–12000 mL·$g_{cat}^{-1}$·$h^{-1}$) was adjusted to maintain a similar CO conversion level (7.2–10.7%) for all studied xZr/Ru catalysts. The gaseous products were analyzed online using a gas chromatograph (Agilent 7890B) equipped with a thermal conductivity detector (TCD) and a flame ionization detector (FID). The liquid product was collected and analyzed by an off-line Shimadzu GC equipped with FID. The calculation of CO conversion, product selectivity, chain-growth probability (α), ruthenium time yield (RuTY), and turnover frequency (TOF) are included as Supplementary Methods. The mass balance, carbon balance, and oxygen balance were calculated and maintained at 100 ± 5%.

### Catalyst characterization

The power X-ray diffraction (XRD) patterns were recorded on a Rigaku Ultima IV X-ray diffractometer equipped with Cu Kα radiation (λ = 1.54056 Å), which was operated at 40 kV and 40 mA with a scanning rate of 2 degrees/min. High-angle annular dark-field scanning transmission electron microscopy (HAADF-STEM) images and the corresponding energy dispersion X-ray analysis (EDX) were conducted on a JEOL JEM-F200 microscope equipped with an Oxford EDX detector. AC-HAADF-STEM was performed on JEM-ARM 300 F with the

EDS measurement. X-ray photoelectron spectroscopy (XPS) data were collected by a ThermoFisher Scientific K-Alpha spectrometer equipped with an Al Kα (1486.6 eV) radiation source. The results were calibrated with the C1s peak at 284.8 eV. The oxidation state of the promoter and the local environments of Zr and Ru were probed by X-ray absorption spectroscopy (XAS) experiments. All data were collected on the BL11B beamline of the Shanghai Synchrotron Radiation Facility (SSRF), China. The Ru K-edge and Zr K-edge spectra were collected at room temperature in fluorescence mode. The experimental methods of XRF, inductively coupled plasma optical emission spectroscopy (ICP-OES), $N_2$ physisorption, TEM, $H_2$-TPR, $H_2$-TPD, CO-TPSR, hydrogen spillover detection by the $WO_3$ powder experiment were supplemented in the Supplementary Methods of the Supplementary Information.

### CO and $H_2$ chemisorption experiments

CO chemisorption experiments were conducted with a Micromeritics 2020 instrument to quantify the exposed surface area and dispersion of the metallic Ru. Therefore, the dispersion of Ru was calculated by assuming the stoichiometry of CO/Ru to be 1/1. Typically, 150 mg of sample was degassed at 300 °C for 4 h. Prior to the chemisorption experiments, the sample was treated in situ at 450 °C in $H_2$ for 2 h and then evacuated. The temperature was decreased to 40 °C under vacuum and then evacuated for 60 min. The adsorbate (CO or $H_2$) was introduced into the system to obtain the uptake of CO or $H_2$.

### In situ time-resolved DRIFTs experiments

The 0Zr/Ru and 0.5Zr/Ru powder samples were filled into the reactor and were pre-reduced in pure $H_2$ (40 mL·$min^{-1}$) at 450 °C for 2 h and cooled with Ar to 260 °C. Then, syngas (CO: $H_2$: Ar = 1:2:7; 50 mL·$min^{-1}$) was introduced into the reactor for 60 min. Simultaneously, in situ, time-resolved DRIFTs were recorded with the time stream on.

### $H_2$/$D_2$ isotopic experiment

A typical effect of the $H_2$/$D_2$ isotopic experiment on the CO conversion and the methane selectivity in CO hydrogenation was observed. The reaction was first operated under $H_2$/CO flow. After the reaction reached a pseudo-steady state, the reactants were switched from $H_2$/CO to $D_2$/CO/Ar. Then, the gas was switched back to the $H_2$ feed. The CO conversion was recorded throughout the entire process.

### $H_2$–$D_2$ exchange experiments

The $H_2$–$D_2$ exchange experiments were carried out in a homemade quartz U-tube reactor system at atmospheric pressure. Fresh sample (0.1 g) was added to the U-tube reactor and then heated at 450 °C for 2 h under a 30 mL·$min^{-1}$ $H_2$ atmosphere. After cooling to 50 °C, the reactor inlet flow was abruptly switched from $H_2$ to $D_2$ (30 mL·$min^{-1}$) via a 4-way valve. After the switch, D atoms react with H atoms on the surface of the sample, resulting in the formation of an HD peak (m/z = 3). After waiting for 10 min, the reactor inlet flow was abruptly switched from $D_2$ to $H_2$ (30 mL·$min^{-1}$). H atoms also react with D atoms on the sample surface, resulting in the formation of an HD peak. The off-gas was continuously monitored by a quadrupole mass spectrometer (INFICON, Transpector CPM).

In situ, diffuse reflectance infrared Fourier transform spectroscopy (DRIFTs) for $H_2$–$D_2$ exchange experiments was performed on a ThermoFisher Scientific FTIR spectrometer (Nicolet iS50) with a liquid-nitrogen-cooled MCT detector. Prior to the measurement, approximately 20 mg of fresh catalyst was reduced in situ at 450 °C for 2 h under a flow of $H_2$ (30 mL·$min^{-1}$), then purged with Ar for 30 min and cooled to 50 °C. The background spectrum (64 scans) with a resolution of 4 $cm^{-1}$ was recorded. Then, the inlet gas was switched to $H_2$ (30 mL·$min^{-1}$) for 30 min and then switched to $D_2$ (30 mL·$min^{-1}$) for 30 min. The spectra were collected by 64 averaged scans with a resolution of 4 $cm^{-1}$.

## Computational methods

All periodic DFT calculations were carried out with the Vienna ab initio simulation package (VASP)[49] using the Perdew-Burke-Ernzerhof (PBE) exchange-correlation functional[50] and the projector-augmented wave (PAW) potentials[51]. An energy cutoff of 400 eV and a Methfessel-Paxton smearing width of 0.20 eV were used. Convergence of the electronic energy in the self-consistent field (SCF) iteration was set to $10^{-4}$ eV, and the forces on all unconstrained atoms were converged to 0.03 eV Å$^{-1}$. The climbing-image nudged elastic band (CI-NEB) method[52] and the improved dimer method (IDM)[53] were carried out to identify the transition states (TS), which were further verified by harmonic vibrational analysis showing a single imaginary mode.

The Ru (001) surface with a $p$ (4 × 4) supercell in four layers was employed in this work, which consists of 64 atoms and a vacuum layer of 15 Å between adjacent slabs along the Z direction. The $ZrO_2$/Ru (001) surface was used to simulate the Zr–O–Ru interface, where a $(ZrO_2)_3$ cluster was placed on the top of the above-mentioned Ru (001) surface. A hydroxylated $ZrO_2$ cluster consisting of $Zr_3O_6H_5$ was further placed on the above Ru (001) surface, which is denoted as the $Zr_3O_6H_5$/Ru (001) surface and is used to simulate the $ZrO_2$-promoted Ru catalyst with a much higher H coverage under the typical reaction conditions. Similar calculations were also performed over the Ru (001) surface at the same H coverage as $Zr_3O_6H_5$/Ru (001). The lattice parameters for the Ru (001), $ZrO_2$/Ru (001), and $Zr_3O_6H_5$/Ru (001) surfaces are a = b = 10.82 and c = 21.42 Å. For these models, the Ru atoms in the bottom two layers were fixed, and the remaining atoms including the $(ZrO_2)_3$ or $Zr_3O_6H_5$ cluster and the adsorbates were allowed to fully relax. The Brillouin zone was sampled with the auto-generated (3 × 3 × 1) Monkhorst-Pack k-point grid.

## Data availability

The data that support the findings of this study are available within the paper and its Supplementary Information. The data generated in this study are provided in the Source Data file. Source data are provided with this paper.

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

## Acknowledgements
This work was financially supported by the Natural Science Foundation of China (U22B20136 and 22293023 received by L.Z., 22072177 received by T.L., 22172188 received by S.L., 22202230 received by Y.A.), the National Key R&D Program of China (2023YFB4103104 received by T.L.), the Natural Science Foundation of Shanghai (22JC1404200 received by L.Z., 21ZR1471700 received by T.L.), Program of Shanghai Academic/Technology Research Leader (20XD1404000 received by L.Z.), and the Youth Innovation Promotion Association of CAS. Specifically, we acknowledge the XAFS station (BL14W1) of the Shanghai Synchrotron Radiation Facility for the XAS test and Shanghai Key Laboratory of High-resolution Electron Microscopy of ShanghaiTech University for the (AC)-HAADF-STEM and STEM-EDS characterization support.

## Author contributions
H.Y. and C.W. conceived the idea. H.Y. was responsible for most of the investigation, methodology development, data collection, and analysis, and writing the original manuscript. C.W. provided assistance in developing most of the experimental methodology and conducting data analysis. Y.W. and F.S. performed the XAS characterization and analysis. X.X. and S.L. performed the DFT calculations. Y.A., T.L., and L.Z. were responsible for the funding and resources acquisition. T.L. and L.Z. were responsible for supervising the project, revising, and editing the manuscript.

## Competing interests
The authors declare no competing interests.
