## [Peer Review File · Nature Communications]

Title Engineering ZrO₂-Ru interface to boost Fischer-Tropsch synthesis to olefinsREVIEWER COMMENTS

Reviewer #1 (Remarks to the Author):

The paper seems to be well-organized to explain the interfacial effects of ZrO₂-Ru metal on the SiO₂ support for FTO reaction. To get rid of some ambiguous points, the following issues should be further discussed to clearly deliver their new-findings as well.

(1) The locations and distribution of ZrO₂ and Ru nanoparticles on the SiO₂ (or ZrO₂ on the Ru nanoparticles) should be double-checked by using additional experiments such as CO chemisorption or XPS(depth profiling). The formation of active sites for H₂ dissociation (spillover) seems to be important to explain an increased activity on the 0.5ZrO₂-Ru-SiO₂ catalyst.

(2) Distributions of active metals on the used catalysts should be included to verify the in-situ reconstruction of metal (oxides) during the FTO reaction. The used catalysts seem to have different morphology as well as dispersion of ZrO₂ and Ru metal, which can be further confirmed by XPS, CO (or H₂) chemisorption as well.

(3) Different product distributions on the different catalysts should be also carefully explained to verify the locations of active metals on the ZrO₂-modified SiO₂ surfaces.

Reviewer #2 (Remarks to the Author):

This manuscript reports a study of the effect of ZrO₂ on the performance of a Na-promoted 2 wt% Ru/SiO₂ catalyst, building on the results reported by this group in ref. 17 for Na-promoted Ru. The most important finding is that the Ru STY increases significantly for a Zr/Ru ratio of 0.5 (4-fold increase), while the CO₂ selectivity decreases but the methane selectivity increases. Other data in Figure 1 convey more or less the same message, but represented in a somewhat different way (conversion at constant GHSV, Ru STY and olefin STY at constant conversion, but different GHSV, TOF at unspecified conditions). The origin of this change in activity is attributed to the formation of ZrO₂/Ru interface sites, but the evidence for such sites is not compelling.

The experiments are performed at somewhat unusual FT conditions of 260C and 5 bar. No data on the catalyst stability are reported, or the change in selectivity with time on stream, or the carbon balance in the experiments.

The SI further shows an important decrease in the O/P ratio with the introduction of ZrO₂, with an important decrease in the olefin selectivity. The data hence point in the direction of modified H₂ adsorption. Since FTS is first order in the hydrogen pressure, an increased rate of hydrogen adsorption (or a higher H* coverage) is expected to increase the activity but decrease the desired olefin selectivity. Evidence for a suggested change in the reaction mechanism is not convincing, HCOH is generally accepted as the key reaction intermediate of the H-assisted mechanism.

The particle size distribution derived from the TEM images is rather broad, with a significant number of very large Ru particles. Such large particles could have a significant effect on the Ru STY since a significant amount of Ru is inactive in the bulk of these large particles.

No post reaction characterization data are included.

CO chemisorption can be quite sensitive to changes in particles size since CO adsorption is partially reversible on Ru particles and hence sensitive to the conditions. H₂ chemisorption could be more sensitive and suggest an increase in H₂ uptake with Zr/Ru ratio (Figure 3). The change in Zr/Ru ratio from XPS is not strong evidence for Zr covering Ru particles. Alternatively, the faster increase in the Zr signal with Zr/Ru ratio of the material could be due to the flatter shape of ZrO₂ particles as compared to the semi-spherical Ru particles. The EXAFS signal for Zr-Ru scattering is not convincing. Did the author try Zr-Si scattering?

The energy profile in Figure 6 shows a 3 eV change in the $\text{CO}^* + \text{H}^* \rightarrow \text{C}^* + \text{OH}^*$ reaction energy when $(\text{ZrO}_2)_3$ islands are introduced, strongly suggesting that ZrO_2 islands are covered with OH^* groups during reaction conditions. The initial state used in the calculations is therefore not a realistic starting point for the reaction, it should be covered by OH^* .

Reviewer #3 (Remarks to the Author):

The authors declared a novel strategy for boosting the catalytic activity toward FTS reaction via constructing ZrO_2 -Ru interface and systematically investigated the relationship between microstructure and catalytic performance. A number of points need clarifying and certain statements require further justification. Some interesting results are presented. However, major revision is needed to further improve the manuscript. For instance, ICP need to further perform to determine the actual content of Ru and Zr. It couldn't be clearly observed that Zr species were mostly located on the surface of Ru NPs rather than on the SiO_2 support as author stated in the manuscript. Although author stated the Ru remain metallic form after catalysis, other obvious diffraction peak in Supplementary Fig. 5 made it hard to judge the state of Ru. Meanwhile, other characterizations such as XPS and XANES should be provided to further determine the state of Ru. The high-resolution scan XPS spectrum of Ru 3p is needed to provide further identify the state of surface Ru. As known, alkali metals usually had great influences on the activity toward FTS. I noticed that Na was added during the catalyst preparation while no relevant data were provided to ensure the similar content and state of Na existed in the catalysts. Moreover, it was inadequate to prove if Na had effects on the ZrO_2 -Ru interface and thus catalytic activity at the present stage. The absent of detailed characterizations on the spent catalyst was also a serious problem for this work. Although author has evidenced the particle size of Ru was almost unchanged, the state of Ru and Zr species after reaction was still unclear, and thus the real active site was also unclear. Furthermore, the stability test should be performed. Besides, there are many English language errors, which should be polished carefully.

Point-by-point responses to all the comments from referees

Reviewer #1

Comments:

The paper seems to be well-organized to explain the interfacial effects of ZrO₂-Ru metal on the SiO₂ support for FTO reaction. To get rid of some ambiguous points, the following issues should be further discussed to clearly deliver their new-findings as well.

Author reply:

Thanks a lot for your valuable comments. The point-by-point responses to your comments are shown as follows:

Comments:

1. The locations and distribution of ZrO₂ and Ru nanoparticles on the SiO₂ (or ZrO₂ on the Ru nanoparticles) should be double-checked by using additional experiments such as CO chemisorption or XPS (depth profiling). The formation of active sites for H₂ dissociation (spillover) seems to be important to explain an increased activity on the 0.5ZrO₂-Ru-SiO₂ catalyst.

Author reply: Thank you very much for your professional comment. The metallic surface area and CO uptake results determined by CO chemisorption for the x Zr/Ru catalysts are displayed in **Fig.1f** and **Fig.3f**, respectively. We found that as the ZrO₂ content increased, both Ru metallic surface area and CO uptake gradually decreased, from 41.4 m²·g_{Ru}⁻¹ and 22.4 μmol·g_{Ru}⁻¹ to 19.0 m²·g_{Ru}⁻¹ and 10.3 μmol·g_{Ru}⁻¹, respectively, suggesting that an excessive amount of ZrO₂ addition would cover the metallic Ru surface.

Fig. 1f. Metallic Ru surface areas (columns) determined by CO chemisorption and TOF (pentagram) plotted against the Zr/Ru molar ratios.

Fig. 3f. Chemisorption uptakes of H₂ and CO over various *x*Zr/Ru catalysts.

Additionally, XPS depth profiling of the 0.5Zr/Ru catalyst was employed to determine the location and distribution of ZrO₂ and Ru nanoparticles on the SiO₂ support, as shown in **Supplementary Fig. 17**. Comparing the XPS intensity of Ru 3*d* and Zr 3*d* with and without etching for 30 s, we observed that the intensity of Ru 3*d* peak increased after the etching, and the calculated relative molar ratio of Zr/Ru decreased from 1.57 to 0.67, which was closer to the nominal value (0.5) of the catalyst. These results suggest that the Zr species are uniformly dispersed on the surface of the 0.5Zr/Ru catalyst, consistent with the results of the (AC)HAADF-STEM-EDS and

XAS characterizations.

As indicated by the reviewer and discussed in the manuscript, the enhanced hydrogen spillover effect and faster hydrogen migration over the 0.5Zr/Ru catalyst than those over the 0Zr/Ru catalyst were verified by additional H₂-D₂ exchange experiments (**Fig. 5a, c**). Moreover, *in situ* DRIFTS for the H₂-D₂ exchange experiments (**Fig. 5b**) and density functional theory (DFT) calculations (**Supplementary Fig. 23**) both confirmed that the ZrO₂-Ru interfacial structure could serve as an H species reservoir and increased the H species coverage near the active center by forming the Zr-OH species. These surface hydroxyls bound to the ZrO₂ promoter could easily be exchanged under hydrogen atmosphere, and the H* species in the surface hydroxyls would actively participate in the H-assisted CO dissociation process leading to higher catalytic activity (**Fig. 6**).

In the revised manuscript, the XPS depth profiling of the reduced 0.5Zr/Ru catalyst was added as *Supplementary Fig. 17 in the Supplementary Information*, and the following discussion was added to the main text:

Supplementary Fig. 17. XPS spectra of the C1s, Ru 3d (a) and Zr 3d (b) levels for the reduced 0.5Zr/Ru catalyst after XPS sputter etching.

“Additionally, Ar-ion sputtering was conducted to strip different depths of the 0.5Zr/Ru catalyst in the XPS measurements (**Supplementary Fig. 17**). The relative Zr/Ru molar ratio decreases from 1.57 to 0.67 as the etching time increases from 0 s to 30 s, which is closer to the nominal value (0.5) of the sample. These observations align

with the results from the (AC)HAADF-STEM-EDS and XAS characterizations, further indicating that the ZrO₂ layer may be homogeneously dispersed on the surface of the 0.5Zr/Ru catalyst.”

2. Distributions of active metals on the used catalysts should be included to verify the in-situ reconstruction of metal (oxides) during the FTO reaction. The used catalysts seem to have different morphology as well as dispersion of ZrO₂ and Ru metal, which can be further confirmed by XPS, CO (or H₂) chemisorption as well.

Author reply: Thank you very much for your valuable comment. We agree with the reviewer that studying the morphology of the used catalyst is important. As shown in the **Supplementary Figs. 7 and 8**, HRTEM images of both the reduced and spent *x*Zr/Ru catalysts presented lattice fringe with a spacing of 2.03 Å, which was assigned to the (101) plane of the hcp metallic Ru phase. Comparison of the average particle sizes of metallic Ru nanoparticles measured by TEM for various reduced and spent samples indicates that adding the ZrO₂ promoter had little influence on the size of metallic Ru NPs. Despite the high-temperature vacuum treatment, CO chemisorption experiment of the used 0.5Zr/Ru catalyst was unsuccessful likely due to the presence of some product species remaining on the catalyst after the reaction. We further supplemented the HAADF-STEM-EDS elemental mapping, XAS, and XPS characterizations of used 0.5Zr/Ru catalyst, as shown in the **Supplementary Fig. 11**, **Supplementary Fig. 13**, and **Supplementary Fig. 15**. These results indicate that there were no significant changes in the distribution and oxidation states of ZrO₂ and metallic Ru after the reaction.

Supplementary Fig. 7. (HR)TEM images and the corresponding particle size distribution for the reduced x Zr/Ru catalysts. (a) 0Zr/Ru, (b) 0.2Zr/Ru, (c) 0.5Zr/Ru, (d) 0.7Zr/Ru, (e) 1Zr/Ru.

Supplementary Fig. 8. (HR)TEM images and the corresponding particle size distribution for the spent x Zr/Ru catalysts. (a) 0Zr/Ru, (b) 0.2Zr/Ru, (c) 0.5Zr/Ru, (d) 0.7Zr/Ru, (e) 1Zr/Ru.

In the revised manuscript, results from the HAADF-STEM-EDS elemental mapping, XAS and XPS characterizations of the used 0.5Zr/Ru catalyst were added as

Supplementary Fig. 11, Supplementary Fig. 13 and Supplementary Fig. 15 in the Supplementary Information and the relevant explanations were added to the main text as following:

Supplementary Fig. 11. HAADF-STEM images and STEM-EDS elemental mapping of the spent 0.5Zr/Ru catalyst. Ru is depicted in green, Si in yellow, Zr in red. The orange arrow represents the line-scanning direction. The related STEM-EDS line-scanning results are displayed in the insets.

“As presented in **Supplementary Fig. 11**, after the reaction the Zr species also displayed no evident aggregation and remained highly dispersed on both the Ru NPs and supports. Besides, the elemental profiles of the STEM-EDS line-scanning further confirmed presence of the Zr species over the Ru NPs.”

Supplementary Fig. 13. (a) Normalized Ru K-edge k_2 -weighted EXAFS spectra of the reduced and spent 0.5Zr/Ru samples. (b) Experimental (colored lines) and best fit (dashed black lines) FT-EXAFS spectra for the reduced and spent 0.5Zr/Ru samples measured at the Zr K-edge.

“The Ru K-edge FT EXAFS spectra of the spent 0.5Zr/Ru also exhibited the metallic Ru phase (**Supplementary Fig. 13a**).”

“The Zr-Ru scattering contributions were also observed in the Zr K-edge EXAFS results of the spent 0.5Zr/Ru catalyst, further demonstrating the strong interaction between the ZrO₂ promoter and metallic Ru resulting in the stable Zr-O-Ru interfacial structures during the reaction (**Supplementary Fig. 13b**).”

Supplementary Fig. 15. XPS spectra of the Ru 3d (a) and Zr 3d (b) line for the spent 0.5Zr/Ru catalyst.

“Chemical states of the Ru and Zr species were found to remain unchanged after the reaction (**Supplementary Fig. 15**). Additionally, the Zr/Ru molar ratio of the spent

0.5Zr/Ru was calculated to be 1.28, close to that of the reduced 0.5Zr/Ru before the reaction. This further confirms that no significant changes occurred to the dispersion of ZrO₂ and metallic Ru during the reaction.”

3. Different product distributions on the doffer catalysts should be also carefully explained to verify the locations of active metals on the ZrO₂-modified SiO₂ surfaces.

Author reply: Thank you very much for your professional comment. ZrO₂ introduction influenced the product distribution, and a volcanic trend was observed for the olefins distribution and the chain-growth probability (α) of hydrocarbons (**Supplementary Fig. 4**) when increasing the Zr/Ru molar ratio. The largest fraction of long-chain olefins (C₅₊) reached 76.7%, which was much higher than that of 0Zr/Ru (57.2%). The α value for 0.5Zr/Ru showed a 1.5-fold increase compared to that of 0Zr/Ru. These results suggested that adding an appropriate amount of ZrO₂ could greatly promote the carbon chain growth.

Supplementary Fig. 4. Olefins distribution (a) and Anderson-Schulz-Flory plots (b) of various x Zr/Ru catalysts under similar CO conversion level ($T = 260$ °C, $H_2/CO = 2$, $P = 5$ bar and $WHSV = 3000 - 12000$ mL \cdot g⁻¹ \cdot h⁻¹).

In accordance with the reviewer’s suggestion, additional discussion regarding the influence of ZrO₂ introduction on the products distribution was added. Based on references 20 and 32 (*Nat. Commun.* 2020, 11, 3185 and *J. Catal.* 2008, 257, 142-151),

we inferred that the reason for the enhanced chain growth capability might be attributed to acceleration of the CO dissociation rate due to the ZrO₂ addition, which increased the coverage of surface CH_x intermediates and enhanced the rate of C-C coupling. Additionally, the increased active hydrogen coverage over the Zr-O-Ru interfacial sites also resulted in the higher CH₄ selectivity.

The relevant discussion was added to the section of “Understanding the enhanced activity of Zr-O-Ru interfacial sites”, as following:

“In addition, the shift in product distribution towards higher-molecular-weight hydrocarbons may be attributed to the enhanced CO activation at Zr-O-Ru interfacial sites, resulting in higher coverages of CH_x species for C-C coupling reactions. A promotional effect toward hydrogenation over Zr-O-Ru interfacial sites also caused a slight increase in CH₄ selectivity^{20,32}.”

Reviewer #2

Comments:

This manuscript reports a study of the effect of ZrO₂ on the performance of a Na-promoted 2 wt% Ru/SiO₂ catalyst, building on the results reported by this group in ref. 17 for Na-promoted Ru. The most important finding is that the Ru STY increases significantly for a Zr/Ru ratio of 0.5 (4-fold increase), while the CO₂ selectivity decreases but the methane selectivity increases. Other data in Figure 1 convey more or less the same message, but represented in a somewhat different way (conversion at constant GHSV, Ru STY and olefin STY at constant conversion, but different GHSV, TOF at unspecified conditions). The origin of this change in activity is attributed to the formation of ZrO₂/Ru interface sites, but the evidence for such sites is not compelling.

Author reply:

Thank you very much for your valuable comments. In this work, we found introducing an appropriate amount of the ZrO₂ promoter significantly could enhance the catalytic activity (CO conversion, RuTY, TOF) and olefins space-time yield compared to the unpromoted catalyst. Experimental determination of the activation energies revealed that the 0.5Zr/Ru catalyst had a significantly reduced apparent activation energy of 44.6 kJ/mol, a substantial 46.1% reduction compared to the 0Zr/Ru catalyst. Combination of structural characterizations and theoretical calculations confirmed that the intimate interaction of the ZrO₂ promoter and metallic Ru enhanced H₂ activation and migration, altered the H-assisted CO dissociation route from the formyl (HCO*) pathway to the hydroxy-methylidyne (COH*) pathway, greatly reduced the energy barrier of C-O bond dissociation, and markedly boosted the intrinsic activity.

Point-by-point responses to your comments are shown as following:

1. The experiments are performed at somewhat unusual FT conditions of 260 °C and 5 bar. No data on the catalyst stability are reported, or the change in selectivity with time on stream, or the carbon balance in the experiments.

Author reply: Thank you very much for your professional comment. As mentioned by the above comment of the present reviewer, this work extends the previous research reported by our group (*Nat. Commun.* 2022, 13, 5987) aiming to improve the utilization

efficiency of the precious metal Ru and focusing on olefins as the target products under the reaction conditions of 260 °C and 5 bar instead of paraffins over the usual FT conditions. The catalyst stability test and the change in product selectivity with time on stream were added as *Supplementary Fig. 3 in the supplementary information*, and the relevant description was added to the main text as following:

Supplementary Fig. 3 Stability test for 0.5Zr/Ru catalyst ($T = 260$ °C, $H_2/CO = 2$, $P = 5$ bar and $WHSV = 3000$ mL·g⁻¹·h⁻¹).

“Performance of 0.5Zr/Ru catalyst remained stable within 110 h of test (**Supplementary Fig. 3**). CO conversion and RuTY remained at the average values of 38.2% and 0.914 mol_{CO}·g_{Ru}⁻¹·h⁻¹, respectively. In addition, CO₂ selectivity remained consistently below 0.5%, while CH₄ selectivity was kept between 5.8% and 7.0%, resulting in C₅₊ product selectivity of approximately 80%.”

Additionally, the mass balance, carbon balance, and oxygen balance were calculated and maintained at 100 ± 5% for the catalytic performance evaluation. We apologize for omitting this information in the experimental section, which was added in the revised manuscript.

“The mass balance, carbon balance, and oxygen balance were calculated and maintained at 100 ± 5%.”

2. The SI further shows an important decrease in the O/P ratio with the introduction of ZrO₂, with an important decrease in the olefin selectivity. The data hence point in the direction of modified H₂ adsorption. Since FTS is first order in the hydrogen pressure, an increased rate of hydrogen adsorption (or a higher H* coverage) is expected to increase the activity but decrease the desired olefin selectivity. Evidence for a suggested change in the reaction mechanism is not convincing, HCOH is generally accepted as the key reaction intermediate of the H-assisted mechanism.

Author reply: Thank you very much for your valuable comment. As the present reviewer has correctly noticed, the introduction of the ZrO₂ promoter boosted the catalytic activity while simultaneously causing a slightly decrease in olefins selectivity. Combining results of the H₂-TPD, H₂ chemisorption, and H₂-D₂ exchange experiments (**Fig. 5**), we found that the ZrO₂ addition significantly affected the abilities of the activation of H₂ and the migration of active H* species. The *in situ* H₂-D₂ exchange DRIFTS experiments further confirmed that the ZrO₂ promoter on metallic Ru surface could capture the H* species to form the Zr-OH species. These active surface hydroxyls might participate in CO activation, which was further confirmed by our DFT calculations. CO dissociation on the (ZrO₂)₃/Ru (001) catalyst model without surface OH* show that the highest energy barrier of H-assisted CO dissociation via the HCO* pathway is similar to that of Ru (001), which is inconsistent with the experimental observation and the experimentally determined apparent activation energy. However, further calculations using a catalyst model with the Zr-OH species showed that the presence of the Zr-OH species could change the reaction pathway for H-assisted CO dissociation, making it faster by forming the COH* intermediate instead of the HCO* intermediate.

According to the reviewer's comments, we further extended our calculations to include the formation of the hydroxy-methylene (HCOH*) intermediate and also the effect of H coverage on the energy barriers.

Using our previous catalyst model, the energy profiles of a more complete reaction pathway for H-assisted CO dissociation including the HCOH* intermediate was added

to the revised *Fig. 6 in the “Computational studies on reaction pathways over Zr-O-Ru interfacial sites”* section and the *Supplementary Fig. 25 and Supplementary Table 6 in the Supplementary Information.*

Fig. 6. DFT optimized structures and reaction pathways studies. Energy profiles and corresponding structures of H-assisted CO dissociation via different intermediates (HCO* and COH*) over metallic Ru catalyst modelled by Ru (001) and the ZrO₂-promoted sample modelled by (ZrO₂)₃/Ru (001). Ru (dark green spheres), Zr (light green spheres), O (red spheres), H (white spheres) and C (gray spheres) atoms are shown.

Supplementary Table 6. Relative energies (eV) of intermediates involved in H-assisted CO dissociation over Ru (001) and (ZrO₂)₃/Ru (001) surfaces.

	Species	Ru (001)	(ZrO ₂) ₃ /Ru (001)
	CO+H	0.00	0.00
COH* Route	COH_Ru	0.66	0.22
	COH_Zr-Ru	-	-0.55
	C+OH	1.26	-1.19
HCO* Route	HCO	1.09	0.71
	HCO+H	0.51	0.17
	HCOH	1.04	0.54
	CO+OH	0.32	0.06

Supplementary Fig. 25. Energy profiles of different intermediates and COH* dissociation over (ZrO₂)₃/Ru (001) surface. Red solid line represents the energy profile of COH* dissociation to C* and OH* pathway and blue solid line represents the energy profile of COH* hydrogenation to HCOH* dissociation pathway.

“Two additional steps are involved for C-O bond dissociation in the HCO* species, hydrogenation of the O atom in HCO* to form the hydroxy-methylene (HCOH*) species with a relatively high energy barrier of $E_a = 1.22$ eV and cleavage of the C-O bond in the CHOH* species with a fairly low energy barrier of $E_a = 0.38$ eV to obtain CH* and OH*. In contrast, over the (ZrO₂)₃/Ru (001) surface, CO dissociation is assisted by the hydroxy species formed at the Zr-O-Ru interfacial structure (i.e. Zr-OH*), which encounters a much lower energy barrier of 0.54 eV and occurs via the hydroxy-methylidyne (COH*) intermediate formed by hydrogen transfer to the O atom in CO*. Although the COH* species adsorbed at the Zr-O-Ru interfacial structure of the ZrO₂-promoted Ru surface can be further hydrogenated to the HCOH* species with a relatively high energy barrier of $E_a = 1.19$ eV, which is comparable to the highest energy barrier for the most favorable pathway of H-assisted CO dissociation on the Ru (001) surface of 1.23 eV, the COH* species can also dissociate with a much lower energy barrier of $E_a = 0.23$ eV (**Supplementary Fig. 25**). Thus, the ZrO₂-promoted Ru catalyst surface considerably accelerates CO dissociation via the COH* pathway assisted by the active Zr-OH* species.”

Furthermore, we performed additional calculations to investigate the effect of H coverage on the metallic Ru surface as well as the Zr-O-Ru interfacial catalyst model on the reaction pathways and energy barriers of H-assisted CO dissociation. The optimized structures of the Zr-O-Ru interfacial structure and pure metallic Ru surface with higher H coverage were shown in the *Supplementary Fig. 26 in the Supplementary Information*, and the calculated energy profiles were added as *Supplementary Fig. 27 in the Supplementary Information*. The computational methods were revised in the experimental section, and the corresponding discussion were added to the main text as following:

“A hydroxylated ZrO₂ cluster consisting of Zr₃O₆H₅ was further placed on the above Ru (001) surface, which is denoted as the Zr₃O₆H₅/Ru (001) surface and is used to simulate the ZrO₂-promoted Ru catalyst with a much higher H coverage under the typical reaction conditions. Similar calculations were also performed over the Ru (001)

surface at the same H coverage as $\text{Zr}_3\text{O}_6\text{H}_5/\text{Ru}$ (001). The lattice parameters for the Ru (001), ZrO_2/Ru (001), and $\text{Zr}_3\text{O}_6\text{H}_5/\text{Ru}$ (001) surfaces are $a = b = 10.82$ and $c = 21.42$ Å. For these models, the Ru atoms in the bottom two layers were fixed, and the remaining atoms including the $(\text{ZrO}_2)_3$ or $\text{Zr}_3\text{O}_6\text{H}_5$ cluster and the adsorbates were allowed to fully relax.”

Supplementary Fig. 26. The top view of optimized structure for Zr-O-Ru interfacial structure (a) and pure metallic Ru surface (b) with higher H coverage. Ru (dark green spheres), Zr (light green spheres), O (red spheres) atoms, and H (white spheres) atoms are shown.

Supplementary Fig. 27. Energy profiles of H-assisted CO dissociation via different intermediates (HCO* and COH*) over Ru (001) and (ZrO₂)₃/Ru (001) with higher H coverage. Red line represents the energy profile of Zr-O-Ru interfacial structure via COH* pathway and blue line represents the energy profile of pure metallic Ru surface via HCO* pathway.

“Considering the higher H coverage on the catalyst surface upon introducing the ZrO₂ promoter, further calculations were performed using models with an increased H coverage (**Supplementary Fig. 26**). The potential energy surface for H-assisted CO dissociation over the catalyst model with the higher H coverage shows that increasing the H coverage can further enhance the promotional effect of ZrO₂ on the H-assisted CO dissociation via the COH* intermediate (**Supplementary Fig. 27**), as the energy barrier for COH* formation assisted by the Zr-OH* species is reduced from 0.54 eV to 0.39 eV, whereas that for COH* dissociation to C* and OH* only slightly increases from 0.23 eV to 0.25 eV. In comparison, increasing the H coverage also accelerates H-assisted CO dissociation on the metallic Ru surface, as the energy barriers for the three elementary steps, namely CO hydrogenation to HCO*, HCO* hydrogenation to

HCOH*, and HCOH* dissociation, are mostly reduced from 1.23 eV, 1.22 eV, and 0.38 eV to 1.17 eV, 1.00 eV and 0.44 eV, respectively. Although increasing the H coverage simultaneously promotes the H-assisted CO dissociation for the COH* route over the ZrO₂-promoted Ru catalyst and the HCO* route over the metallic Ru surface, our theoretical calculations confirm that the ZrO₂ promoter considerably facilitates CO dissociation via the COH* pathway assisted by the active Zr-OH* species during high H coverage.”

3. The particle size distribution derived from the TEM images is rather broad, with a significant number of very large Ru particles. Such large particles could have a significant effect on the Ru STY since a significant amount of Ru is inactive in the bulk of these large particles.

Author reply: Thank you very much for your professional comment. The particle size distributions of all the *x*Zr/Ru catalysts range from 2 to 11 nm, with the average particle sizes between 5.1 and 6.2 nm (**Supplementary Fig. 7**). Thus, there is no significant change in the average particle size with the addition of the ZrO₂ promoter (**Supplementary Fig. 8**). Hence, in this work, we believe that the variation in RuTY is not mainly due to changes in the particle size, but primarily due to the introduction of the ZrO₂ promoter. Furthermore, the intrinsic activity (TOF) of the catalysts was also used as a benchmark to investigate the influence of the ZrO₂ promoter on the catalytic activity of the Ru/SiO₂ catalysts, and an 8.5-fold higher TOF was observed over the 0.5Zr/Ru catalyst compared with the unpromoted catalyst (**Fig. 1f**).

Supplementary Fig. 7. (HR)TEM images and the corresponding particle size distribution for the reduced x Zr/Ru catalysts. (a) 0Zr/Ru, (b) 0.2Zr/Ru, (c) 0.5Zr/Ru, (d) 0.7Zr/Ru, (e) 1Zr/Ru.

Supplementary Fig. 8. (HR)TEM images and the corresponding particle size distribution for the spent x Zr/Ru catalysts. (a) 0Zr/Ru, (b) 0.2Zr/Ru, (c) 0.5Zr/Ru, (d) 0.7Zr/Ru, (e) 1Zr/Ru.

Fig. 1f. Metallic Ru surface areas (columns) determined by CO chemisorption and TOF (pentagram) plotted against the Zr/Ru molar ratios.

Additionally, it is well known that particle size can significantly influence the catalytic performance, and in this work we further investigated the size effect of Ru-based catalysts in the Fischer-Tropsch synthesis to olefins reaction (*Figure 1 (for response)*). We found that the TOF was rapidly reduced as the Ru particle size decreases below 4 nm, while it remained a higher value and kept almost unchanged as the size increases above 4 nm. This trend is very similar to the effect of Co particle size in the FTS process (*J. Am. Chem. Soc. 2006, 128, 3956*). In this work, most of the Ru nanoparticles for the samples with or without the ZrO₂ promoter fall in the range of 4~10 nm. Therefore, the Ru size effect on the TOF could be reasonably excluded, and the observed quite different catalytic behavior could be mostly ascribed to the effect of the ZrO₂ promoter.

Figure 1 (for response) The size effect of Ru-based catalyst in the Fischer-Tropsch synthesis to olefins reaction (**unpublished data**).

4. No post reaction characterization data are included.

Author reply: Thank you very much for your valuable suggestions. In the supplementary information, we provided XRD and (HR)TEM characterization results for the spent catalysts, as shown in **Supplementary Figs. 6c and 8**. These results indicate that the catalysts remained in the metallic Ru phase after the reaction, and there was no notable change in the average particle size of Ru nanoparticles.

Supplementary Fig. 6c. XRD patterns for the spent catalysts. (▲ represents the diffraction peak of diluent SiO₂.)

Supplementary Fig. 8. (HR)TEM images and the corresponding particle size distribution for the spent x Zr/Ru catalysts. (a) 0Zr/Ru, (b) 0.2Zr/Ru, (c) 0.5Zr/Ru, (d) 0.7Zr/Ru, (e) 1Zr/Ru.

Additionally, we performed HAADF-STEM-EDS, XAS, and XPS characterizations for the spent 0.5Zr/Ru catalyst, and further analyzed its structural changes before and after the reaction. Results from these characterizations were added as *Supplementary Figs. 11, 13 and 15 in the Supplementary Information*, and the corresponding discussion was added to the main text as following:

Supplementary Fig. 11. HAADF-STEM images and STEM-EDS elemental mapping of the spent 0.5Zr/Ru catalyst. Ru is depicted in green, Si in yellow, Zr in red. The orange arrow represents the line-scanning direction. The related STEM-EDS line-scanning results are displayed in the insets.

“As presented in **Supplementary Fig. 11**, after the reaction the Zr species also displayed no evident aggregation and remained highly dispersed on both the Ru NPs and supports. Besides, the elemental profiles of the STEM-EDS line-scanning further confirmed presence of the Zr species over the Ru NPs.”

Supplementary Fig. 13. (a) Normalized Ru K-edge k_2 -weighted EXAFS spectra of the reduced and spent 0.5Zr/Ru samples. (b) Experimental (colored lines) and best fit (dashed black lines) FT-EXAFS spectra for the reduced and spent 0.5Zr/Ru samples

measured at the Zr K-edge.

“The Ru K-edge FT EXAFS spectra of the spent 0.5Zr/Ru also exhibited the metallic Ru phase (**Supplementary Fig. 13a**).”

“The Zr-Ru scattering contributions were also observed in the Zr K-edge EXAFS results of the spent 0.5Zr/Ru catalyst, further demonstrating the strong interaction between the ZrO₂ promoter and metallic Ru resulting in the stable Zr-O-Ru interfacial structures during the reaction (**Supplementary Fig. 13b**).”

Supplementary Fig. 15. XPS spectra of the Ru 3d (a) and Zr 3d (b) line for the spent 0.5Zr/Ru catalyst.

“Chemical states of the Ru and Zr species were found to remain unchanged after the reaction (**Supplementary Fig. 15**). Additionally, the Zr/Ru molar ratio of the spent 0.5Zr/Ru was calculated to be 1.28, close to that of the reduced 0.5Zr/Ru before the reaction. This further confirms that no significant changes occurred to the dispersion of ZrO₂ and metallic Ru during the reaction.”

5. CO chemisorption can be quite sensitive to changes in particles size since CO adsorption is partially reversible on Ru particles and hence sensitive to the conditions. H₂ chemisorption could be more sensitive and suggest an increase in H₂ uptake with Zr/Ru ratio (Figure 3). The change in Zr/Ru ratio from XPS is not strong evidence for Zr covering Ru particles. Alternatively, the faster increase in the Zr signal with Zr/Ru ratio of the material could be due to the flatter shape of ZrO₂ particles as compared to the semi-spherical Ru particles. The EXAFS signal

for Zr-Ru scattering is not convincing. Did the author try Zr-Si scattering?

Author reply: We greatly appreciate your professional comment. We agree with the reviewer that CO and H₂ chemisorption are sensitive to the particle size of the catalysts. Many researchers employed CO and H₂ chemisorption to measure the particle size and dispersion of Ru active sites (*Nat Commun.*, 2020, 11, 3185; *J. Catal.* 1992, 137, 212; *J. Catal.* 1973, 28,289). We compared the average particle sizes of metallic Ru nanoparticles by TEM for all the reduced and spent *x*Zr/Ru samples, which indicates that adding the ZrO₂ promoter had little influence on the particle size of metallic Ru NPs (**Supplementary Figs. 7 and 8**). Therefore, the Ru particle size effect on CO chemisorption could be reasonably ignored. According to the CO chemisorption results (**Fig. 1f and 3f**), we observed a gradual decrease in the CO uptake with an increase in the Zr/Ru molar ratio, accompanied by a corresponding decrease in the exposed Ru metallic surface area. These results suggest that the added ZrO₂ would cover the metallic Ru surface.

Fig. 1f. Metallic Ru surface areas (columns) determined by CO chemisorption and TOF (pentagram) plotted against the Zr/Ru molar ratios.

Fig. 3f. Chemisorption uptakes of H₂ and CO over various x Zr/Ru catalysts.

The XPS results alone cannot conclusively demonstrate that ZrO₂ covers metallic Ru nanoparticles, so we further combined the (AC)HAADF-STEM-EDS and XAS characterization results (**Fig. 2c-h** and **Fig. 3b-c**) to demonstrate that ZrO₂ is amorphous and highly dispersed, and formation of ZrO₂ particles was not observed. The additional STEM-EDS line-scanning and XPS depth profiles further confirm that ZrO₂ is uniformly dispersed on the metallic Ru and SiO₂ support. These characterization results were added as *Supplementary Figs. 11 and 17 in the Supplementary Information*, and the corresponding discussion was added to the main text as following:

Supplementary Fig. 11. HAADF-STEM images and STEM-EDS elemental mapping of the spent 0.5Zr/Ru catalyst. Ru is depicted in green, Si in yellow, Zr in red. The orange arrow represents the line-scanning direction. The related STEM-EDS line-scanning results are displayed in the insets.

“As presented in **Supplementary Fig. 11**, after the reaction the Zr species also displayed no evident aggregation and remained highly dispersed on both the Ru NPs and supports. Besides, the elemental profiles of the STEM-EDS line-scanning further confirmed presence of the Zr species over the Ru NPs.”

Supplementary Fig. 17. XPS spectra of the C1s, Ru 3d (a) and Zr 3d (b) levels for the reduced 0.5Zr/Ru catalyst after XPS sputter etching.

“Additionally, Ar-ion sputtering was conducted to strip different depths of the 0.5Zr/Ru catalyst in the XPS measurements (**Supplementary Fig. 17**). The relative Zr/Ru molar ratio decreases from 1.57 to 0.67 as the etching time increases from 0 s to 30 s, which is closer to the nominal value (0.5) of the sample. These observations align with the results from the (AC)HAADF-STEM-EDS and XAS characterizations, further indicating that the ZrO₂ layer may be homogeneously dispersed on the surface of the 0.5Zr/Ru catalyst.”

Regarding the Zr-Ru scattering in the Zr K-edge EXAFS signal, Zr-Si as the fitting model was initially considered. However, the fitting parameters of Zr-Si scattering was unreasonable, whereas introducing Zr-Ru scattering resulted in a much better fitting. Comparison of the fitting results using Zr-Si and Zr-Ru scatterings is shown in *Figure 2 (for response)*. Despite reasonable fitting parameters, we were unable to obtain results that match the original data. Additionally, due to the weaker electron scattering ability of Si atoms compared to Ru atoms, it is very challenging to observe any coordination peaks in the R space for Si atoms. In contrast, the possibility of Ru to form Zr-Ru scattering is much higher, consistent with our fitting results.

Figure 2 (for response) Experimental (colored lines) and fit (dashed black lines) FT-EXAFS spectra using Zr-Si and Zr-Ru scattering for the reduced 0.5Zr/Ru sample measured at the Zr K-edge.

6. The energy profile in Figure 6 shows a 3 eV change in the $\text{CO}^* + \text{H}^* \rightarrow \text{C}^* + \text{OH}^*$ reaction energy when $(\text{ZrO}_2)_3$ islands are introduced, strongly suggesting that ZrO_2 islands are covered with OH^* groups during reaction conditions. The initial state used in the calculations is therefore not a realistic starting point for the reaction, it should be covered by OH^* .

Author reply: We greatly appreciate your valuable comment. In our calculations of the reaction pathways, we actually utilized the H^* in the $\text{ZrO}_2\text{-OH}^*$ species as the reactant, and this is now clearly labeled in *the revised vision of the Fig. 6*, where the reactant CO adsorbed over the $(\text{ZrO}_2)_3/\text{Ru}$ (001) surface reacts with the H atom adsorbed at the O site (H_{O}^*) to obtain the hydroxy-methylidyne (COH^*) intermediate, and the ZrO_2 islands can be considered to have a low H coverage.

Fig. 6. DFT optimized structures and reaction pathways studies. Energy profiles and corresponding structures of H-assisted CO dissociation via different intermediates (HCO^* and COH^*) over metallic Ru catalyst modelled by Ru (001) and the ZrO_2 -

promoted sample modelled by $(\text{ZrO}_2)_3/\text{Ru}$ (001). Ru (dark green spheres), Zr (light green spheres), O (red spheres), H (white spheres) and C (gray spheres) atoms are shown.

Based on your suggestions, we augmented the catalyst models to have a higher H coverage over the ZrO_2 islands or the metallic Ru surface, as shown in *Supplementary Fig. 26 in the Supplementary Information*. The corresponding energy profile of H-assisted CO dissociation via the different intermediates (HCO^* and COH^*) over the Ru (001) surface covered with 5 H atoms and the $\text{Zr}_3\text{O}_6\text{H}_5/\text{Ru}$ (001) surface were given as *Supplementary Fig. 27 the Supplementary Information*. Our computational results show that when the ZrO_2 islands are covered with more OH^* groups in the initial state, CO dissociation over the $\text{Zr}_3\text{O}_6\text{H}_5/\text{Ru}$ (001) surface encounters an even lower energy barrier of $E_a = 0.39$ eV for the hydrogenation of CO^* to COH^* compared with that over the $(\text{ZrO}_2)_3/\text{Ru}$ (001) surface ($E_a = 0.54$ eV). Meanwhile, HCO^* formation over the Ru (001) surface covered with 5 H atoms involved a considerably higher energy barrier of $E_a = 1.17$ eV. Thus, the ZrO_2 -promoted Ru catalyst surface greatly accelerates CO dissociation via the COH^* pathway assisted by the active Zr-OH^* species.

Supplementary Fig. 26. The top view of optimized structure for Zr-O-Ru interfacial structure (a) and pure metallic Ru surface (b) with higher H coverage. Ru (dark green spheres), Zr (light green spheres), O (red spheres) atoms, and H (white spheres) atoms are shown.

Supplementary Fig. 27. Energy profiles of H-assisted CO dissociation via different intermediates (HCO* and COH*) over Ru (001) and (ZrO₂)₃/Ru (001) with higher H coverage. Red line represents the energy profile of Zr-O-Ru interfacial structure via COH* pathway and blue line represents the energy profile of pure metallic Ru surface via HCO* pathway.

Reviewer #3

Comments:

The authors declared a novel strategy for boosting the catalytic activity toward FTS reaction via constructing ZrO₂-Ru interface and systematically investigated the relationship between microstructure and catalytic performance. A number of points need clarifying and certain statements require further justification. Some interesting results are presented. However, major revision is needed to further improve the manuscript.

Author reply:

Thank you very much for your valuable comments. The point-by-point responses to your comments are shown as following:

Comments:

1. For instance, ICP need to further perform to determine the actual content of Ru and Zr.

Author reply: Thank you very much for your valuable suggestions. We further performed ICP-OES experiments to obtain the actual content of Ru and Zr elements as presented in **Supplementary Table 2**. The Ru content in a series of $x\text{Zr}/\text{Ru}$ samples closely approached the nominal value (approximately 2 wt.%), while the Zr content as indicated by its molar ratio relative to Ru was also close to the theoretical value.

In the revised manuscript, the ICP-OES results of the $x\text{Zr}/\text{Ru}$ catalysts were presented in the *Supplementary Table 2 in the Supplementary Information*, and the corresponding experimental method was revised as following:

“The element content of various samples was measured by an energy dispersive X-ray fluorescence (XRF, Rigaku ZSX Primus II) and ICP-OES (Varian ICP-OES 720).”

Supplementary Table 2. Elemental composition of various fresh catalysts determined by XRF and ICP-OES.

Catalyst	Ru content (wt.%) ^a	Zr content (wt.%) ^a	Na content (wt.%) ^b	Ru content (wt.%) ^b	Zr content (wt.%) ^b	Zr/Ru molar ratio ^b
0Zr/Ru	1.86	-	0.27	1.93	-	-
0.2Zr/Ru	1.84	0.31	0.25	1.90	0.41	0.24
0.5Zr/Ru	1.87	0.82	0.26	2.00	0.91	0.51
0.7Zr/Ru	1.90	1.24	0.27	2.10	1.59	0.84
1Zr/Ru	1.98	1.93	0.38	1.87	2.03	1.24

a: Determined by XRF. b: Determined by ICP-OES.

2. It couldn't be clearly observed that Zr species were mostly located on the surface of Ru NPs rather than on the SiO₂ support as author stated in the manuscript.

Author reply: Thank you very much for your professional comment. We agree with the reviewer that evidence for the Zr species to be mostly located on the surface of Ru NPs is not sufficiently robust. Therefore, we removed the statement “and it also seems that the Zr species were mostly located on the surface of Ru NPs rather than on the SiO₂ support” in the revised manuscript.

3. Although author stated the Ru remain metallic form after catalysis, other obvious diffraction peak in Supplementary Fig. 5 made it hard to judge the state of Ru. Meanwhile, other characterizations such as XPS and XANES should be provided to further determine the state of Ru. The high-resolution scan XPS spectrum of Ru 3p is needed to provide further identify the state of surface Ru.

Author reply: We greatly appreciate for your valuable comment. We apologize for not providing detailed description of the extra diffraction peaks of the spent *x*Zr/Ru catalysts in **Supplementary Fig. 6c**. We added a legend indicating that the additional diffraction peaks in Supplementary Fig. 6c can be attributed to the residual diluent SiO₂. The diffraction peaks located at $2\theta = 38.4^\circ$, 42.2° and 44.0° correspond to the (100),

(002) and (101) crystal planes of metallic Ru, respectively, as indicated in the PDF#06-0663 card.

In the revised Supplementary Information, *Supplementary Fig. 6* was revised as following:

Supplementary Fig. 6. (a) N_2 adsorption-desorption isotherms for various $x\text{Zr/Ru}$ samples; (b) XRD patterns for the fresh catalysts; (c) XRD patterns for the spent catalysts. (\blacktriangle represents the diffraction peak of diluent SiO_2 .)

Additionally, XANES and XPS measurements were also employed to characterize the state of Ru for the $x\text{Zr/Ru}$ catalysts, as shown in **Fig. 2b**, **Supplementary Fig. 12**, and **Fig. 3d**. According to the reviewer's suggestion, XPS spectra of Ru 3p of the $x\text{Zr/Ru}$ catalysts were fitted and added as *Supplementary Fig. 16 in the Supplementary*

Information, with the additional discussion provided in the revised main text.

Fig. 2b. Ru K-edge EXAFS spectra of the reduced $x\text{Zr/Ru}$ samples.

Supplementary Fig. 12. Ru K-edge XANES spectra of the reduced $x\text{Zr/Ru}$ samples.

Fig. 3d. XPS spectra of various reduced catalysts at the Ru 3d level with Ru 3d and C 1s contributions.

In the revised manuscript, several sentences were further revised for better clarity in *Paragraph 4 and 7 of “Identification of the Zr-O-Ru interfacial structure” section*, as following:

“We further investigated the local environment of the Ru sites in the Zr-promoted catalysts by X-ray adsorption spectroscopy (XAS). According to the XANES spectra at the Ru K-edge, the Ru near-edge of the reduced samples almost coincides with that of Ru foil, suggesting the existence of metallic Ru sites (Supplementary Fig. 12).”

“From the XPS spectra for Ru 3d and Ru 3p shown in Fig. 3d and Supplementary Fig. 16, the binding energies of the Ru 3d_{5/2} peak and the Ru 3d_{3/2} peak were measured to be 279.9 – 280.2 eV and 461.0 – 461.5 eV, respectively, suggesting that Ru is in the metallic state.”

Supplementary Fig. 16. XPS spectra of various reduced catalysts at the Ru 3p level.

4. As known, alkali metals usually had great influences on the activity toward FTS. I noticed that Na was added during the catalyst preparation while no relevant data were provided to ensure the similar content and state of Na existed in the catalysts. Moreover, it was inadequate to prove if Na had effects on the ZrO₂-Ru interface and thus catalytic activity at the present stage.

Author reply: Thank you very much for your valuable comment. To investigate the influence of the ZrO₂-Ru interfacial structure on the performance of Ru-based catalysts, the Na/Ru molar ratio was fixed at 0.5 during the catalyst preparation to eliminate the impact of the Na promoter. According to the reviewer's suggestion, ICP-OES measurements was performed to obtain the Na content of all the *x*Zr/Ru catalysts, as presented in the *Supplementary Table 2 in the Supplementary Information*. It was observed that all the catalysts maintained a similar level of Na content. To further elucidate the state of Na present in the catalysts, the Na element mapping was determined by STEM-EDS for all the reduced *x*Zr/Ru samples, as shown in *Supplementary Figs. 9 and 10 in the Supplementary Information*. The Na elements are found to be uniformly dispersed.

Supplementary Table 2. Elemental composition of various fresh catalysts determined by XRF and ICP-OES.

Catalyst	Ru content (wt.%) ^a	Zr content (wt.%) ^a	Na content (wt.%) ^b	Ru content (wt.%) ^b	Zr content (wt.%) ^b	Zr/Ru molar ratio ^b
0Zr/Ru	1.86	-	0.27	1.93	-	-
0.2Zr/Ru	1.84	0.31	0.25	1.90	0.41	0.24
0.5Zr/Ru	1.87	0.82	0.26	2.00	0.91	0.51
0.7Zr/Ru	1.90	1.24	0.27	2.10	1.59	0.84
1Zr/Ru	1.98	1.93	0.38	1.87	2.03	1.24

a: Determined by XRF. **b:** Determined by ICP-OES.

Supplementary Fig. 9. Na elemental mapping of the reduced 0.5Zr/Ru sample determined by STEM-EDS.

Supplementary Fig. 10. HAADF-STEM images and STEM-EDS elemental mapping of the reduced (a) 0Zr/Ru, (b) 0.2Zr/Ru and (c) 1Zr/Ru catalysts. Ru is depicted in green, Si in yellow, Zr in red, and Na in purple.

Regarding the impact of Na on the catalytic performance as noted by the reviewer, we have previously conducted detailed studies on the influence of the Na promoter on the Fischer-Tropsch synthesis performance of Ru-based catalysts (*Nat Commun.* 2022

13, 5987). The addition of the Na promoter altered the electronic structure and surface microenvironment of metallic Ru nanoparticles. Changes in the local electronic structure and the decreased reactivity of chemisorbed H species on metallic Ru surfaces tailor the reaction pathway to favor olefins production, resulting in up to 80% olefins selectivity with a Na/Ru molar ratio of 0.5. In this work, while maintaining a fixed Na/Ru molar ratio of 0.5, we investigated the significant enhancement of the catalytic activity in Fischer-Tropsch synthesis to olefins reaction by varying the content of the ZrO₂ promoter in the Ru-based catalysts. We found that the highly dispersed Zr species could strongly bind the metallic Ru NPs to form Zr-O-Ru interface sites. Furthermore, the increased coverage of the local H species and the reactivity around Zr-O-Ru interfaces sites altered the H-assisted CO dissociation route from formyl (HCO*) pathway to the hydroxy-methylidyne (COH*) pathway, greatly reduced the energy barrier of C-O dissociation, and markedly boosted the intrinsic activity.

5. The absent of detailed characterizations on the spent catalyst was also a serious problem for this work. Although author has evidenced the particle size of Ru was almost unchanged, the state of Ru and Zr species after reaction was still unclear, and thus the real active site was also unclear.

Author reply: Thank you very much for your professional comment. In the supplementary information, we provided XRD and (HR)TEM characterization results of the spent catalysts, as shown in **Supplementary Figs. 6c and 8**. These results indicate that the catalysts remained in the metallic Ru phase after the reaction, and there was no notable change in the average particle size of Ru nanoparticles.

Supplementary Fig. 6c. XRD patterns for the spent catalysts. (▲ represents the diffraction peak of diluent SiO₂.)

Supplementary Fig. 8. (HR)TEM images and the corresponding particle size distribution for the spent x Zr/Ru catalysts. (a) 0Zr/Ru, (b) 0.2Zr/Ru, (c) 0.5Zr/Ru, (d) 0.7Zr/Ru, (e) 1Zr/Ru.

Additionally, the states of the Ru and Zr species after the reaction were further determined by HAADF-STEM-EDS, XAS, and XPS characterizations.

In the revised manuscript, these characterization results of the spent 0.5Zr/Ru catalyst were added as *Supplementary Figs. 11, 13 and 15 in the Supplementary Information*, and the corresponding discussion was added to the *“Identification of the*

Zr-O-Ru interfacial structure” section as following:

Supplementary Fig. 11. HAADF-STEM images and STEM-EDS elemental mapping of the spent 0.5Zr/Ru catalyst. Ru is depicted in green, Si in yellow, Zr in red. The orange arrow represents the line-scanning direction. The related STEM-EDS line-scanning results are displayed in the insets.

“As presented in **Supplementary Fig. 11**, after the reaction the Zr species also displayed no evident aggregation and remained highly dispersed on both the Ru NPs and supports. Besides, the elemental profiles of the STEM-EDS line-scanning further confirmed presence of the Zr species over the Ru NPs.”

Supplementary Fig. 13. (a) Normalized Ru K-edge k_2 -weighted EXAFS spectra of the reduced and spent 0.5Zr/Ru samples. (b) Experimental (colored lines) and best fit (dashed black lines) FT-EXAFS spectra for the reduced and spent 0.5Zr/Ru samples measured at the Zr K-edge.

“The Ru K-edge FT EXAFS spectra of the spent 0.5Zr/Ru also exhibited the metallic Ru phase (**Supplementary Fig. 13a**).”

“The Zr-Ru scattering contributions were also observed in the Zr K-edge EXAFS results of the spent 0.5Zr/Ru catalyst, further demonstrating the strong interaction between the ZrO₂ promoter and metallic Ru resulting in the very stable Zr-O-Ru interfacial structures during the reaction (**Supplementary Fig. 13b**).”

Supplementary Fig. 15. XPS spectra of the Ru 3d (a) and Zr 3d (b) line for the spent 0.5Zr/Ru catalyst.

“Chemical states of the Ru and Zr species were found to remain unchanged after the reaction (**Supplementary Fig. 15**). Additionally, the Zr/Ru molar ratio of the spent

0.5Zr/Ru was calculated to be 1.28, close to that of the reduced 0.5Zr/Ru before the reaction. This further confirms that no significant changes occurred to the dispersion of ZrO₂ and metallic Ru during the reaction.”

6. Furthermore, the stability test should be performed.

Author reply: We greatly appreciate for your professional comment. Catalytic stability test of the 0.5Zr/Ru catalysts were performed and presented in **Supplementary Fig. 3**. The 0.5Zr/Ru catalyst showed high stability within 110 h of test. CO conversion and RuTY were maintained at around 38% and 0.9 mol_{CO}·g_{Ru}⁻¹·h⁻¹, respectively, with less than 0.5% selectivity towards CO₂ and 5.8% to 7.0% selectivity for CH₄.

In the revised manuscript, results for the stability test of the 0.5Zr/Ru catalyst were added as *Supplementary Fig. 3 in the Supplementary Information*, and the corresponding discussion was added to the main text as following:

Supplementary Fig. 3 Stability test for 0.5Zr/Ru catalyst (T = 260 °C, H₂/CO = 2, P = 5 bar and WHSV = 3000 mL·g⁻¹·h⁻¹).

“Performance of 0.5Zr/Ru catalyst remained stable within 110 h of test (**Supplementary Fig. 3**). CO conversion and RuTY remained at the average values of 38.2% and 0.914 mol_{CO}·g_{Ru}⁻¹·h⁻¹, respectively. In addition, CO₂ selectivity remained consistently below 0.5%, while CH₄ selectivity was kept between 5.8% and 7.0%, resulting in C₅₊ product selectivity of approximately 80%.”

7. Besides, there are many English language errors, which should be polished carefully.

Author reply: Thank you very much for your valuable comments. We are very sorry for some of the grammatical errors and typos in our previous manuscript, we have gone through the entire manuscript and carefully checked and improved the English writing in the revised manuscript.

REVIEWER COMMENTS

Reviewer #1 (Remarks to the Author):

The paper seems to be well revised to meet the referees' comments, and I believe the paper can be accepted at it is.

Reviewer #2 (Remarks to the Author):

The authors have addressed my comments and have greatly improved and expanded the experimental data in the manuscript. While the effect of the ZrO₂ promotor on the activity and selectivity of Ru/Na/SiO₂ is interesting, and already seen at Zr/Ru ratios of 0.2, evidence for Ru/Zr interface sites and for a change in reaction mechanism are less conclusive and both the title and the abstract in my opinion overstate this interpretation.

The evidence for Ru-ZrO₂ interface sites is somewhat strengthened and uses the Zr K-edge EXAFS, CO chemisorption and XPS depth profiling. It is plausible that some ZrO₂ is close to the Ru particles. The role and effect of Na remains unclear. Experiments with Zr/Ru without Na would be instructive. The EXAFS data in Figure 2(for reponse) seem incorrect - the experimental data for the same 0.5Zr/Ru sample (colored lines) are different.

Evidence for the change in reaction mechanism comes from H₂-D₂ isotope studies and from DFT calculations. The isotope studies support the formation of OH groups and the enhanced H₂ activation on 0.5Zr/Ru.

The DFT calculations in Figure 6 show an almost 3 eV difference between C* + OH* on Ru and on (ZrO₂)₃/Ru. OH* hence binds very strongly to the (ZrO₂)₃ cluster to make this reaction 3 eV more favorable. This indicates that the (ZrO₂)₃ model cluster is not a stable structure during reaction and will bind multiple OH*. The DFT calculations are less convincing.

Reviewer #3 (Remarks to the Author):

The author has systematically characterized the spent catalysts and clarified the effect of Na in the catalysts. I recommend that the article can be accepted after minor revision.

1. The activity is better to calculate based on ICP-results.
2. The Method section should be supplemented to coincide with the content in revised manuscript.
3. In the background section, it would be more interesting to summarize the related attempts on the recent development of related catalysts. It would be necessary to discuss by referring some articles, such as: ACS Catalysis, 2016, 6: 100-114.; ACS Catalysis, 2020, 10: 7894-7906.; Nature Communication, 2020, 11: 3185.; Journal of the American Chemical Society, 2023, 145: 7113–7122.
4. The relevant experimental should be performed to explore if Zr could promote activity significantly without Na.
5. The resolution of Fig. S11 should be further optimized.

Point-by-point response to the reviewers' comments

Reviewer #1

Comments:

The paper seems to be well revised to meet the referees' comments, and I believe the paper can be accepted at it is.

Author reply:

We would like to thank Referee #1 for the positive feedback on our revised manuscript. We also sincerely thank Referee #1 for the helpful comments and suggestions, which greatly helped to improve the quality of this manuscript.

Reviewer #2

Comments:

The authors have addressed my comments and have greatly improved and expanded the experimental data in the manuscript.

Author reply:

Thank you very much for your positive feedback and professional comments. The point-by-point responses to your comments are shown as follows:

Comments:

1. While the effect of the ZrO₂ promotor on the activity and selectivity of Ru/Na/SiO₂ is interesting, and already seen at Zr/Ru ratios of 0.2, evidence for Ru/Zr interface sites and for a change in reaction mechanism are less conclusive and both the title and the abstract in my opinion overstate this interpretation. The evidence for Ru-ZrO₂ interface sites is somewhat strengthened and uses the Zr K-edge EXAFS, CO chemisorption and XPS depth profiling. It is plausible that some ZrO₂ is close to the Ru particles.

Author reply: Thank you for your professional comments. In this work, we utilized various characterizations techniques, including Zr K-edge EXAFS, CO chemisorption, and XPS depth profiling, to provide evidence that the highly dispersed ZrO₂ promoter strongly binds with the Ru nanoparticles, forming the Zr-O-Ru interfacial structure. By combing H₂-D₂ isotope studies and DFT calculations, it can be inferred that the formation of the Zr-O-Ru interfacial structure benefits hydrogen migration, and accelerates hydrogen spillover. More specifically, with the assistance from the generated hydroxyl species, the route for H-assisted CO dissociation was changed from the formyl (HCO*) pathway to the hydroxy-methylidyne (COH*) pathway. This alteration significantly reduced the energy barrier of CO dissociation, resulting in substantially higher reactivity, which is further elaborated in our responses to Comment #4 of the present referee.

2. The role and effect of Na remains unclear. Experiments with Zr/Ru without Na would be instructive.

Author reply: Thank you very much for your valuable comment. Regarding the role and effect of Na on the catalytic performance as noted by the reviewer, we have previously conducted detailed studies on the influence of the Na promoter on the Fischer-Tropsch synthesis performance of Ru-based catalysts (*Nat. Commun.* 2022 13, 5987). Generally, the supported metallic Ru catalyst without Na promoter is an outstanding FTS catalyst for producing saturated paraffins. The addition of Na promoter alters the electronic structure and surface microenvironment of metallic Ru nanoparticles, leading to the reduced reactivity of the chemisorbed H species on metallic Ru surfaces to favor the reaction pathway for olefins production with olefins selectivity of up to 80% at a Na/Ru molar ratio of 0.5.

In this work, while maintaining a fixed Na/Ru molar ratio of 0.5, we investigated the significant enhancement of the catalytic activity for Fischer-Tropsch to olefins by varying the content of the ZrO₂ promoter over the Ru-based catalysts. We found that the highly dispersed Zr species could strongly bind the metallic Ru nanoparticles to form Zr-O-Ru interface sites. Furthermore, with the assistance from the hydroxyl species adsorbed on the Zr sites, the Zr-O-Ru interfaces sites altered the H-assisted CO dissociation route from formyl (HCO*) pathway to the hydroxy-methylidyne (COH*) pathway, greatly reducing the energy barrier of C-O dissociation, and markedly boosting the intrinsic activity.

Additionally, we prepared Ru/SiO₂ and Zr-Ru/SiO₂ catalysts without Na promoter using impregnation method, denoted as Ru/SiO₂-0Na and Zr-Ru/SiO₂-0Na. The Ru loading was maintained at 2 wt.% with the same Zr/Ru molar ratio of 0.5. Catalytic performance under the same reaction conditions was shown in *Figure 1 (for response)*. It was found that the addition of Zr promoter notably enhanced the catalytic activity, as evidenced by the increase of CO conversion from 24.5% for Ru/SiO₂-0Na to 45.4% for Zr-Ru/SiO₂-0Na

under the same reaction conditions (220 °C, 1 MPa, 3000 h⁻¹, and H₂/CO=2). Moreover, it moderately improves CH₄ selectivity from 1.8% to 4.4% and decreases C₅₊ selectivity from 94.1% to 90.5%. Due to the low water-gas shift activity of Ru-based catalysts, CO₂ selectivity remains almost negligible for both Ru/SiO₂-0Na and Zr-Ru/SiO₂-0Na catalysts.

Figure 1 (for response) Effect of the Zr promoter on the catalytic performance without Na under the same reaction conditions (220 °C, 1 MPa, 3000 h⁻¹, and H₂/CO=2).

XRD patterns of the reduced catalysts without Na promoter were shown in *Figure 2 (for response)*. It can be observed that adding the Zr promoter does not affect the phase of the catalysts, which remains metallic Ru (JCPDS, 06-0663). Furthermore, no diffraction peaks for the crystalline phase of the ZrO₂ promoter were observed. Comparison of the average particle sizes of metallic Ru nanoparticles measured by TEM for all reduced samples indicates that the ZrO₂ loading had little influence on the size of metallic Ru nanoparticles (*Figure 3 (for response)* and *Table 1 (for response)*). The dispersion of metallic Ru nanoparticles, calculated based on the average particle size obtained from the TEM results, is determined to be 16.7% and 17.7% for Ru/SiO₂-0Na and Zr-Ru/SiO₂-0Na, respectively. The intrinsic activity was further calculated as displayed in *Figure 1 (for response)* with 0.092 s⁻¹ and 0.161 s⁻¹ for Ru/SiO₂-

0Na and Zr-Ru/SiO₂-0Na, respectively. It is evident that introducing the Zr promoter significantly enhances the intrinsic activity of the Ru/SiO₂ catalyst.

The above experimental results suggest that the promotional effect of Zr on supported Ru-based catalysts with or without the Na promoter are similar. In this work, we focus on the promotional effect of Zr promoter on Fischer-Tropsch synthesis to olefins, and the Na/Ru molar ratio was fixed at 0.5 for all the studied samples.

Figure 2 (for response) XRD patterns for the reduced catalysts of Ru/SiO₂-0Na and Zr-Ru/SiO₂-0Na.

Figure 3 (for response) TEM images and the corresponding particle size distribution for the reduced catalysts. (a, c) Ru/SiO₂-0Na, (b, d) Zr-Ru/SiO₂-0Na.

Table 1 (for response) Crystallite size, particle size and dispersion of metallic Ru nanoparticles of the reduced Ru/SiO₂-0Na and Zr-Ru/SiO₂-0Na catalysts.

Catalysts	Crystallite size (nm) ^a	Particle size (nm) ^b	D _{TEM} ^b (%)
Ru/SiO ₂ -0Na	8.8	6.7	16.7
Zr-Ru/SiO ₂ -0Na	9.1	6.3	17.7

^a Calculated by XRD. ^b Determined by TEM.

3. The EXAFS data in Figure 2 (for response) seem incorrect - the experimental data for the same 0.5Zr/Ru sample (colored lines) are different.

Author reply: Thank you very much for your valuable suggestion. We apologize for the confusion regarding the data. The incorrect data has been replaced, as shown in *Figure 4 (for response)*. It was observed that the fitting parameters of Zr-Si scattering was unreasonable, whereas introducing Zr-Ru scattering resulted in a much better fitting.

Figure 4 (for response) Experimental (colored lines) and fitted (dashed black lines) FT-EXAFS spectra using Zr-Si and Zr-Ru scatterings for the reduced 0.5Zr/Ru sample measured at the Zr K-edge.

4. Evidence for the change in reaction mechanism comes from H₂-D₂ isotope studies and from DFT calculations. The isotope studies support the formation of OH groups and the enhanced H₂ activation on 0.5Zr/Ru. The DFT calculations in Figure 6 show an almost 3 eV difference between C* + OH* on Ru and on (ZrO₂)₃/Ru. OH* hence binds very strongly to the (ZrO₂)₃ cluster to make this reaction 3 eV more favorable. This indicates that the (ZrO₂)₃ model cluster is not a stable structure during reaction and will bind multiple OH*. The DFT calculations are less convincing.

Author reply: Thank you very much for your constructive feedback and professional comment. According to the reviewer's comments, we have

performed additional calculations using the $\text{Zr}_3\text{O}_6\text{H}_5/\text{Ru}$ (001) surface as the primary model for simulating the ZrO_2 -promoted Ru catalysts during the reaction, as shown in Supplementary Fig. 25a in the Supplementary Information. The detailed computational results are shown in Fig. 6 and Supplementary Table 6 in the Supplementary Information. The energy profiles of COH^* dissociation via different intermediates over the $\text{Zr}_3\text{O}_6\text{H}_5/\text{Ru}$ (001) surface are displayed in Supplementary Fig.26 in the Supplementary Information. Additionally, the adsorption energies of OH^* at the different sites on the Ru (001) and $\text{Zr}_3\text{O}_6\text{H}_5/\text{Ru}$ (001) surfaces are shown in Supplementary Table 7 in the Supplementary Information. The energy profiles for removing OH^* species in the form of H_2O molecules and regenerating the Zr sites were also calculated, as shown in Supplementary Fig. 27 in the Supplementary Information.

Due to the facile activation of H_2 , easy migration of the resulting H^* adsorbate to the ZrO_2 promoter, and the strong binding of the OH^* species at the Zr site, we used the $\text{Zr}_3\text{O}_6\text{H}_5/\text{Ru}$ (001) surface as the primary model for the ZrO_2 -promoted Ru catalyst. The strong binding of the OH^* species at the Zr site significantly lowers the overall potential energy surface and makes C-O bond breaking kinetically and thermodynamically more favorable. Further hydrogenation of the above OH^* species enables the regeneration of both the Ru and Zr sites after the H-assisted CO dissociation step. On the Ru (001) surface, hydrogenation of the OH^* species incurs a relatively high energy barrier of 1.22 eV, which is reduced to 1.13 eV for hydrogenation of the OH^* species at the Zr site, suggesting that the $\text{Zr}_3\text{O}_6\text{H}_5/\text{Ru}$ (001) surface is easier to regenerate than the Ru (001) surface.

The corresponding explanations were revised in the main text as following:

Supplementary Fig. 25. The top view of optimized structure for Zr-O-Ru interfacial structure (a) and pure metallic Ru surface (b) with higher H coverage. Ru (dark green spheres), Zr (light green spheres), O (red spheres) atoms, and H (white spheres) atoms are shown.

Fig. 6. DFT predictions of surface structures and reaction pathways. Energy profiles and corresponding transition state structures of H-assisted CO

dissociation via different intermediates (HCO^* and COH^*) over metallic Ru catalyst modelled by Ru (001) and the ZrO_2 -promoted sample modelled by $\text{Zr}_3\text{O}_6\text{H}_5/\text{Ru}$ (001). The adsorption sites other than the Ru site are indicated and further denoted by subscripts. Ru (dark green spheres), Zr (light green spheres), O (red spheres), H (white spheres) and C (gray spheres) atoms are shown.

Supplementary Table 6. Relative energies (eV) of intermediates involved in H-assisted CO dissociation over Ru (001) and $\text{Zr}_3\text{O}_6\text{H}_5/\text{Ru}$ (001) surfaces.

	Species	Ru (001)	Species	$\text{Zr}_3\text{O}_6\text{H}_5/\text{Ru}$ (001)
COH* Route	CO+H	0.00	CO+H_o	0.00
	COH	0.66	COH	-0.12
	C+OH	1.26	COH_Zr-Ru	-0.35
			C+OH_Zr	-0.78
HCO* Route	HCO	1.09	HCO	0.09
	HCO+H	0.51	HCO+H	-0.39
	HCOH	1.04	HCOH	0.32
	CH+OH	0.32	CH+OH	-0.42

Supplementary Fig. 26. Energy profiles of different intermediates and COH^* dissociation over $\text{Zr}_3\text{O}_6\text{H}_5/\text{Ru}$ (001) surface. Red solid line represents the energy profile of COH^* dissociation to C^* and OH^* pathway and blue solid

line represents the energy profile of COH* hydrogenation to HCOH* dissociation pathway.

Supplementary Table 7. Adsorption energy (eV) of the OH* species over the Ru (001) and Zr₃O₆H₅/Ru (001) surfaces with respect to gas phase H₂O and H₂.

Site	E _{ads, OH} /eV
The Ru site of Ru (001)	-0.26
The Zr site of Zr ₃ O ₆ H ₅ /Ru (001)	-1.33

Supplementary Fig. 27. Energy profiles for the hydrogenation of OH* species to H₂O molecules and the further desorption of H₂O molecules at the Ru sites of Ru (001) (top) and the Zr sites of Zr₃O₆H₅/Ru (001) (bottom).

“Under typical reaction conditions, H₂ activation is enhanced, and facile migration of the H* adsorbate after introducing the ZrO₂ promoter further results in hydroxylation of the (ZrO₂)₃ island and provides abundant Zr-OH species. Thus, a Zr₃O₆H₅ cluster was used to investigate the role of the Zr-OH species on CO dissociation under actual reaction conditions (**Supplementary Fig. 25a**). The potential energy surface for H-assisted CO dissociation over the Ru (001) and Zr₃O₆H₅/Ru (001) surfaces was further shown in **Fig. 6 and Supplementary Table 6**. Our calculations show that the Ru (001) and Zr₃O₆H₅/Ru (001) surfaces exhibit distinct H-assisted CO activation mechanisms involving different intermediates species⁵⁰. Over the Ru (001) surface, H-assisted CO dissociation via the formyl (HCO*) route (E_a = 1.23 eV) is much faster than that via the hydroxy-methylidyne (COH*) intermediates (E_a = 1.74 eV). Two additional steps are involved for C-O bond dissociation in the HCO* species, hydrogenation of the O atom in HCO* to form the hydroxy-methylene (HCOH*) species with a relatively high energy barrier of E_a = 1.22 eV and cleavage of the C-O bond in the CHOH* species with a fairly low energy barrier of E_a = 0.38 eV to obtain CH* and OH*. In contrast, over the Zr₃O₆H₅/Ru (001) surface, CO dissociation is assisted by the hydroxy species formed at the Zr-O-Ru interfacial structure (i.e. Zr-OH*), which encounters a much lower energy barrier of 0.39 eV and occurs via the hydroxy-methylidyne (COH*) intermediate formed by hydrogen transfer to the O atom in CO*. Although the COH* species adsorbed at the Zr-O-Ru interfacial structure of the ZrO₂-promoted Ru surface can be further hydrogenated to the HCOH* species with a relatively high energy barrier of E_a = 1.15 eV, which is comparable to the highest energy barrier for the most favorable pathway of H-assisted CO dissociation on the Ru (001) surface of 1.23 eV, the COH* species can also dissociate with a much lower energy barrier of E_a = 0.25 eV (**Supplementary Fig. 26**). Thus, the ZrO₂-promoted Ru catalyst surface considerably accelerates CO dissociation via the COH* pathway assisted by the active Zr-OH* species. Our DFT calculations suggest that the OH* species are strongly adsorbed at the

Zr sites with an adsorption energy of $E_{\text{ads}} = -1.33$ eV with gas phase H_2O and H_2 as the reference states, much stronger than those adsorbed at the Ru sites with $E_{\text{ads}} = -0.26$ eV (**Supplementary Table 7**), thus the Zr atoms can stabilize the OH^* species better than Ru atoms. The strong binding of OH^* species at the Zr site significantly lower the overall potential energy surface and makes the C-O bond dissociation kinetically and thermodynamically more favorable. On the Ru (001) surface, hydrogenation of OH^* species is slightly endothermic by 0.39 eV with a high energy barrier of 1.22 eV, and further desorption of the adsorbed H_2O molecules is modestly endothermic by 0.44 eV (**Supplementary Fig. 27**). In comparison, hydrogenation of OH^* species adsorbed at the Zr sites encounters a slightly lower energy barrier of 1.13 eV, with an endothermicity of 0.81 eV, and further desorption of the adsorbed H_2O molecules to recover the Zr sites are more endothermic on the $\text{Zr}_3\text{O}_6\text{H}_5/\text{Ru}$ (001) surface ($E_{\text{des}} = 1.08$ eV), consistent with the high stability of the Zr-OH species on the ZrO_2 -promoted Ru catalysts, although H_2O desorption from the Zr site should readily occur considering the typical reaction temperatures.”

Furthermore, to better understand the effect of H coverage on the reactivity of the ZrO_2 -Ru interfacial sites, DFT calculations were also performed using the H-covered Ru (001) surface with the same H coverage as $\text{Zr}_3\text{O}_6\text{H}_5/\text{Ru}$ (001) is shown in Supplementary Fig. 25b in the Supplementary Information. Comparison of the HCO^* route over the Ru (001) and H-covered Ru (001) surfaces is added in Supplementary Fig. 28 in the Supplementary Information. For comparison, the energy profiles of H-assisted CO dissociation via different routes over the $(\text{ZrO}_2)_3/\text{Ru}$ (001) surface, which may serve as a model for ZrO_2 -promoted Ru catalysts under low H_2 pressure, are given in Supplementary Fig. 29 in the Supplementary Information. Our DFT calculations show that increasing the surface H coverage can also accelerate the H-assisted CO dissociation step for the preferred HCO^* route. Furthermore, compared with the $\text{Zr}_3\text{O}_6\text{H}_5/\text{Ru}$ (001) surface, the $(\text{ZrO}_2)_3/\text{Ru}$ (001) surface is less favorable for H-

assisted CO dissociation due to the higher energy barrier for the formation of the COH* intermediate. Thus, the effect of the H coverage appears to be similar for the Ru catalysts with or without the ZrO₂ promoter.

The corresponding explanations was revised in the main text as following:

Supplementary Fig. 28. Energy profiles of H-assisted CO dissociation via the HCO* route over the Ru (001) and H-covered Ru (001) surfaces.

Supplementary Fig. 29. Energy profiles of H-assisted CO dissociation via different intermediates (HCO* and COH*) over the (ZrO₂)₃/Ru (001) surface.

“Although the ZrO_2 promoter on the metallic Ru surface could better stabilize the H^* adsorbates, the Ru surface may also have some adsorbed H^* species especially under high H coverage. The potential energy surface for H-assisted CO dissociation over the H-covered Ru catalyst model shows that increasing the H coverage could also promote the H-assisted CO dissociation for the preferred HCO^* route (**Supplementary Figs. 25b and 28**), as the energy barriers for the three elementary steps, namely CO hydrogenation to HCO^* , HCO^* hydrogenation to HCOH^* , and HCOH^* dissociation, are mostly reduced from 1.23 eV, 1.22 eV, and 0.38 eV to 1.17 eV, 1.00 eV and 0.44 eV, respectively. DFT calculations were also performed using the $(\text{ZrO}_2)_3/\text{Ru}$ (001) model, which can be considered to model the ZrO_2 -promoted Ru catalyst under much lower H_2 pressure (**Supplementary Figs. 22 and 29**). The energy barrier for COH^* formation assisted by the Zr-OH^* species increases from 0.39 eV for the $\text{Zr}_3\text{O}_6\text{H}_5/\text{Ru}$ (001) model to 0.54 eV for the $(\text{ZrO}_2)_3/\text{Ru}$ (001) model, whereas that for COH^* dissociation to C^* and OH^* only slightly decreases from 0.25 eV to 0.23 eV, indicating the hydroxylated ZrO_2 promoter considerably enhances CO dissociation via the COH^* pathway assisted by the Zr-OH^* species under high H coverage.”

Reviewer #3

Comments:

The author has systematically characterized the spent catalysts and clarified the effect of Zr in the catalysts. I recommend that the article can be accepted after minor revision.

Author reply:

Thank you very much for your timely feedback and valuable comments. The point-by-point responses to your comments are shown as follows:

1. The activity is better to calculate based on ICP-results.

Author reply: Thank you very much for your valuable suggestion. We recalculated the reaction rate (RuTY), olefins space-time yield (STY), and turnover frequency (TOF) of the various $x\text{Zr/Ru}$ catalysts based on the ICP results.

In the revised manuscript, *Fig. 1b, Fig. 1d, Fig. 1f in the Fig. 1 and Supplementary Fig. 2 in the supplementary Information* were revised, and the corresponding description was also revised as following:

Fig. 1. CO hydrogenation performance. (a, b) CO conversion and RuTY, (c) product selectivity, and (d) space-time yield of olefins product measured over ZrO₂-promoted SiO₂-supported ruthenium catalysts. (e) Apparent activation energies for CO conversion determined by the Arrhenius plots shown in **Supplementary Fig. 5**. (f) Metallic Ru surface areas (columns) determined by CO chemisorption and TOF (pentagram) plotted against the Zr/Ru molar ratios.

Supplementary Fig. 2. The comparison of olefins STY over 0.5Zr/Ru and 5%Ru/SiO₂ catalysts at similar CO conversion level (T = 260 °C, P = 0.5 MPa, H₂/CO=2, and WHSV = 3000 – 12000 mL·g_{cat}⁻¹·h⁻¹).

“Herein, we show that the ZrO₂-Ru interface could be engineered by loading the ZrO₂ promoter onto silica-supported Ru nanoparticles (ZrRu/SiO₂), achieving 7.6 times higher intrinsic activity and ~45% reduction in the apparent activation energy compared with the unpromoted Ru/SiO₂ catalyst.”

“In this work, ZrO₂-doping strategy is used to successfully engineer silica-supported Ru nanoparticles to form the Zr-O-Ru interfacial structure (ZrRu/SiO₂), which shows 7.6 times increase in the Ru-normalized intrinsic activity and a 4.1-fold increase in the space-time-yield of olefin products.”

“The 0.5Zr/Ru sample exhibited the highest RuTY value of 0.955 mol_{CO}·g_{Ru}⁻¹·h⁻¹ and the TOF value of 0.320 s⁻¹, which were 5.7 and 7.6 times higher than that of the unpromoted sample (0.167 mol_{CO}·g_{Ru}⁻¹·h⁻¹ and 0.04 s⁻¹), respectively (Fig. 1b, 1f). However, further increase in the ZrO₂ content caused the catalytic activity to decrease, resulting in a RuTY value of 0.504 mol_{CO}·g_{Ru}⁻¹·h⁻¹ for 1Zr/Ru, which was still 3.0-fold higher than that of the 0Zr/Ru catalyst (Fig. 1b).”

“Although a slight decrease in olefins selectivity was observed with the increasing Zr/Ru molar ratio, the space-time yield (STY) of olefins gradually increased and reached a maximum value of 493.7 g·mol_{Ru}⁻¹·h⁻¹ for 0.5Zr/Ru,

which was 4.1-fold higher than that of 0Zr/Ru (**Fig. 1d**). As the Zr/Ru molar ratio further increased, the STY of olefins decreased to 287.9 g·mol_{Ru}⁻¹·h⁻¹ for 0.7Zr/Ru and 298.3 g·mol_{Ru}⁻¹·h⁻¹ for 1Zr/Ru, which remained far higher than that of 0Zr/Ru.”

“Compared with the unpromoted 0Zr/Ru sample, a 7.6-fold higher TOF for CO conversion (0.320 s⁻¹) and a 4.1-fold higher space-time-yield of olefins (493.7 g·mol_{Ru}⁻¹·h⁻¹) were obtained for 0.5Zr/Ru.”

2. The Method section should be supplemented to coincide with the content in revised manuscript.

Author reply: Thank you very much for your valuable comment. We apologize for the confusion caused by some experimental methods being included in the Supplementary Information. We have added relevant explanations to *the Methods section* as follows:

“The experimental methods of XRF, ICP-OES, N₂ physisorption, TEM, H₂-TPR, H₂-TPD, CO-TPSR, hydrogen spillover detection by the WO₃ powder experiment were supplemented in the **Supplementary Methods** in the **Supplementary Information**.”

3. In the background section, it would be more interesting to summarize the related attempts on the recent development of related catalysts. It would be necessary to discuss by referring some articles, such as: ACS Catalysis, 2016, 6: 100-114.; ACS Catalysis, 2020, 10: 7894-7906.; Nature Communication, 2020, 11: 3185.; Journal of the American Chemical Society, 2023, 145: 7113–7122.

Author reply: Thank you very much for your professional suggestion. The articles mentioned by the reviewer are indeed highly relevant and informative for our work. We have cited “Nature Communication, 2020, 11: 3185.” as “Ref. 20”, “ACS Catalysis, 2020, 10: 7894-7906.” as “Ref. 24”, and “Journal of the American Chemical Society, 2023, 145: 7113–7122.” as “Ref. 25” in

Paragraph 3 of the Introduction section. The introduction section was also revised based on the reviewer's suggestion.

In the revised manuscript, *Paragraph 3 of the Introduction section* has been revised as following:

“Interface engineering has emerged as an effective strategy to tune the catalytic behavior of FTS. By constructing abundant interfacial sites on metal nanoparticles, e.g., metal-oxide sites¹⁹⁻²², single atom-decorated surface sites²³⁻²⁵, the geometric and electronic properties as well as the coordination environment of metal NPs could be substantially changed, leading to much improved intrinsic activity and product selectivity. These newly formed interfacial active sites can facilitate the activation of the reactants and promote the formation and transformation of the intermediates. **For instance, the Co-Zr interface has been successfully engineered to promote CO adsorption and dissociation, thereby enhancing the reactivity of Co-based FTS catalysts. Liu et al. suggested that the Co-ZrO₂ interface, featuring single-site dispersion of ZrO₂ on surfaces of Co nanoparticles, enhanced both H₂ adsorption and CO dissociation²⁴. Li et al. reported that the Ru1Zr1/Co catalyst with dual atomic sites of Ru and Zr on surfaces of Co nanoparticles, effectively weakened the C–O bond and promoted C-C coupling²⁵.** However, the role of interfacial active sites in alternating the reaction pathway and the evolution of activated CO species have been rarely reported.”

4.The relevant experimental should be performed to explore if Zr could promote activity significantly without Na.

Author reply: Thank you very much for your professional comment. We further prepared Ru/SiO₂ and Zr-Ru/SiO₂ catalysts without adding the Na promoter using the impregnation method, denoted as Ru/SiO₂-0Na and Zr-Ru/SiO₂-0Na. The Ru loading was maintained at 2 wt.% with the same Zr/Ru molar ratio of 0.5. Catalytic performance under the same reaction conditions was shown in *Figure 1 (for response)*. It was found that the addition of Zr promoter notably

enhanced the catalytic activity, as evidenced by the increase of CO conversion from 24.5% for Ru/SiO₂-0Na to 45.4% for Zr-Ru/SiO₂-0Na under the same reaction conditions (220 °C, 1 MPa, 3000 h⁻¹, and H₂/CO=2). Moreover, it moderately improves CH₄ selectivity from 1.8% to 4.4% and decreases C₅₊ selectivity from 94.1% to 90.5%. Due to the low water-gas shift activity of Ru-based catalysts, CO₂ selectivity remains almost negligible for both Ru/SiO₂-0Na and Zr-Ru/SiO₂-0Na.

Figure 1 (for response) Effect of Zr promoter on catalytic performance without Na under the same reaction conditions (220 °C, 1 MPa, 3000 h⁻¹, and H₂/CO=2).

XRD patterns of the reduced catalysts without Na promoter were shown in *Figure 2 (for response)*. It can be observed that adding the Zr promoter does not affect the phase of the catalysts, which remains metallic Ru (JCPDS, 06-0663). Furthermore, no diffraction peaks for the crystalline phase of the ZrO₂ promoter were observed. Comparison of the average particle sizes of metallic Ru nanoparticles measured by TEM for all reduced samples indicates that the ZrO₂ loading had little influence on the size of metallic Ru nanoparticles (*Figure 3 (for response)* and *Table 1 (for response)*). The dispersion of metallic Ru nanoparticles, calculated based on the average particle size obtained from the TEM results, is determined to be 16.7% and 17.7% for Ru/SiO₂-0Na and Zr-Ru/SiO₂-0Na, respectively. The intrinsic activity was further calculated as

displayed in *Figure 1 (for response)* with 0.092 s^{-1} and 0.161 s^{-1} for Ru/SiO₂-0Na and Zr-Ru/SiO₂-0Na, respectively. It is evident that introducing the Zr promoter significantly enhances the intrinsic activity of the Ru/SiO₂ catalyst.

The above experimental results suggest that the promotional effect of Zr on supported Ru-based catalysts with or without the Na promoter are similar. In this work, we focus on the promotional effect of Zr on Fischer-Tropsch synthesis to olefins, and the Na/Ru molar ratio was fixed at 0.5 for all the studied samples.

Figure 2 (for response) XRD patterns for the reduced catalysts for Ru/SiO₂-0Na and Zr-Ru/SiO₂-0Na.

Figure 3 (for response) TEM images and the corresponding particle size distribution for the reduced catalysts. (a, c) Ru/SiO₂-0Na, (b, d) Zr-Ru/SiO₂-0Na.

Table 1 (for response) Crystallite size, particle size and dispersion of metallic Ru nanoparticles of the reduced Ru/SiO₂-0Na and Zr-Ru/SiO₂-0Na catalysts.

Catalysts	Crystallite size (nm) ^a	Particle size (nm) ^b	D _{TEM} ^b (%)
Ru/SiO ₂ -0Na	8.8	6.7	16.7
Zr-Ru/SiO ₂ -0Na	9.1	6.3	17.7

^a Calculated by XRD. ^b Determined by TEM.

5.The resolution of Fig. S11 should be further optimized.

Author reply: Thank you very much for your valuable suggestion. We apologize for the insufficient resolution of Fig. S11 due to variations in the test batches

and image adjustments. We have significantly improved resolution of the images for the distribution of the Zr element.

Supplementary Fig. 11. HAADF-STEM images and STEM-EDS elemental mapping of the spent 0.5Zr/Ru catalyst. Ru is depicted in green, Si in yellow, Zr in red. The orange arrow represents the line-scanning direction. The related STEM-EDS line-scanning results are displayed in the insets.

REVIEWERS' COMMENTS

Reviewer #2 (Remarks to the Author):

The authors have again improved the manuscript. The Na-free experiments are an important addition. I would appreciate if these experiments are included in the SI of the manuscript, not only in the response file. The DFT calculations have been significantly improved.

Reviewer #3 (Remarks to the Author):

The author adopted various characterizations and theoretical calculations to reveal the formed Zr-O-Ru interfacial structure, which could strengthen the hydrogen spillover effect and serves as a reservoir for active H species by forming Zr-OH* species. Particularly, the ZrO₂ could change reaction route from HCO* to COH* which significantly lowered the energy barrier of rate-limiting CO dissociation step and greatly increased the reactivity. This work is benefit for design efficient industrial Fisher-Tropsch synthesis catalysts, and thus I recommend it publish on Nature Communication.

Point-by-point response to the reviewers' comments

Reviewer #2

Comments:

The authors have again improved the manuscript. The Na-free experiments are an important addition. I would appreciate if these experiments are included in the SI of the manuscript, not only in the response file. The DFT calculations have been significantly improved.

Author reply:

Thank you very much for your positive feedback and professional comments. The Na-free experiments results have been included in the revised Supplementary Information, and corresponding descriptions have been added to the main text.

“To exclude the effect of sodium and demonstrate the promoting effect of ZrO₂ promoter on Ru-based catalysts in the traditional Fischer-Tropsch synthesis process, we prepared and evaluated the catalytic performance of Na-free Ru/SiO₂-0Na and Zr-Ru/SiO₂-0Na. As shown in Supplementary Fig. 6, it was found that the addition of Zr promoter notably enhanced the catalytic activity, as evidenced by the increase of CO conversion from 24.5% for Ru/SiO₂-0Na to 45.4% for Zr-Ru/SiO₂-0Na. The above experimental results suggest that the promotional effect of Zr on supported Ru-based catalysts with or without the Na promoter are similar. Moreover, powder X-ray diffraction (XRD) patterns and transmission electron microscopy (TEM) images present similar metallic Ru phases without ZrO₂ phase and similar particle sizes of metallic Ru nanoparticles (Supplementary Figs. 7, 8). In this work, we focus on the promotional effect of Zr promoter on Fischer-Tropsch synthesis to olefins, therefore the promotional effect of ZrO₂ will be studied in detail over xZr/Ru catalysts.”

Additionally, the detailed notes were supplemented in the Supplementary Information.

Supplementary Fig. 6. Effect of the Zr promoter on the catalytic performance of Na-free catalysts under the same reaction conditions (220 °C, 1 MPa, 3000 h⁻¹, and H₂/CO=2).

Notes:

It was found that the addition of Zr promoter notably enhanced the catalytic activity, as evidenced by the increase of CO conversion from 24.5% for Ru/SiO₂-0Na to 45.4% for Zr-Ru/SiO₂-0Na under the same reaction conditions (220 °C, 1 MPa, 3000 h⁻¹, and H₂/CO=2). Moreover, the addition of Zr moderately improves CH₄ selectivity from 1.8% to 4.4% and decreases C₅₊ selectivity from 94.1% to 90.5%. Due to the low water-gas shift activity of Ru-based catalysts, CO₂ selectivity remains almost negligible for both Ru/SiO₂-0Na and Zr-Ru/SiO₂-0Na catalysts. The intrinsic activity was further calculated as displayed in Supplementary Fig. 6 with 0.092 s⁻¹ and 0.161 s⁻¹ for Ru/SiO₂-0Na and Zr-Ru/SiO₂-0Na, respectively. It is evident that introducing the Zr promoter significantly enhances the intrinsic activity of the Ru/SiO₂ catalyst.

Supplementary Fig. 7. XRD patterns for the reduced catalysts of Ru/SiO₂-0Na and Zr-Ru/SiO₂-0Na.

Notes:

XRD patterns of the reduced catalysts without Na promoter were shown in Supplementary Fig. 7. It can be observed that adding the Zr promoter does not affect the phase of the catalysts, which remains metallic Ru (JCPDS, 06-0663). Furthermore, no diffraction peaks for the crystalline phase of the ZrO₂ promoter were observed.

Supplementary Fig. 8. TEM images and the corresponding particle size distribution for the Na-free reduced catalysts. (a, c) Ru/SiO₂-0Na, (b, d) Zr-Ru/SiO₂-0Na.

Notes:

Comparison of the average particle sizes of metallic Ru nanoparticles measured by TEM for all reduced samples indicates that the ZrO₂ loading exhibits little influence on the size of metallic Ru nanoparticles (Supplementary Fig. 8). The dispersion of metallic Ru nanoparticles, calculated based on the average particle size obtained from the TEM results, is determined to be 16.7% and 17.7% for Ru/SiO₂-0Na and Zr-Ru/SiO₂-0Na, respectively.

And the descriptions of catalyst preparation were added in the Methods Section.

“Ru/SiO₂-0Na and Zr-Ru/SiO₂-0Na catalysts without the Na promoter were also prepared by the impregnation method. The Ru loading was maintained at 2 wt.% with the same Zr/Ru molar ratio of 0.5.”

Reviewer #3

Comments:

The author adopted various characterizations and theoretical calculations to reveal the formed Zr-O-Ru interfacial structure, which could strengthen the hydrogen spillover effect and serves as a reservoir for active H species by forming Zr-OH* species. Particularly, the ZrO₂ could change reaction route from HCO* to COH* which significantly lowered the energy barrier of rate-limiting CO dissociation step and greatly increased the reactivity. This work is benefit for design efficient industrial Fisher-Tropsch synthesis catalysts, and thus I recommend it publish on Nature Communication.

Author reply:

We would like to thank Referee #3 for the positive feedback on our revised manuscript. We also sincerely thank Referee #3 for the helpful comments and suggestions, which greatly helped to improve the quality of this manuscript.